# Portable, bedside, low-field magnetic resonance imaging for evaluation of intracerebral hemorrhage

Mercy H. Mazurek [1,9], Bradley A. Cahn[1,9], Matthew M. Yuen[1], Anjali M. Prabhat[1], Isha R. Chavva[1], Jill T. Shah[1], Anna L. Crawford[1], E. Brian Welch[2], Jonathan Rothberg[2], Laura Sacolick[2], Michael Poole[2], Charles Wira[3], Charles C. Matouk [4], Adrienne Ward[5], Nona Timario[5], Audrey Leasure[1], Rachel Beekman[1], Teng J. Peng[1], Jens Witsch [1], Joseph P. Antonios [4], Guido J. Falcone[1], Kevin T. Gobeske[1], Nils Petersen[1], Joseph Schindler[1], Lauren Sansing[1], Emily J. Gilmore[1], David Y. Hwang[1], Jennifer A. Kim[1], Ajay Malhotra[6], Gordon Sze[6], Matthew S. Rosen [7], W. Taylor Kimberly [8✉] & Kevin N. Sheth [1✉]

Radiological examination of the brain is a critical determinant of stroke care pathways. Accessible neuroimaging is essential to detect the presence of intracerebral hemorrhage (ICH). Conventional magnetic resonance imaging (MRI) operates at high magnetic field strength (1.5–3 T), which requires an access-controlled environment, rendering MRI often inaccessible. We demonstrate the use of a low-field MRI (0.064 T) for ICH evaluation. Patients were imaged using conventional neuroimaging (non-contrast computerized tomography (CT) or 1.5/3 T MRI) and portable MRI (pMRI) at Yale New Haven Hospital from July 2018 to November 2020. Two board-certified neuroradiologists evaluated a total of 144 pMRI examinations (56 ICH, 48 acute ischemic stroke, 40 healthy controls) and one ICH imaging core lab researcher reviewed the cases of disagreement. Raters correctly detected ICH in 45 of 56 cases (80.4% sensitivity, 95%CI: [0.68–0.90]). Blood-negative cases were correctly identified in 85 of 88 cases (96.6% specificity, 95%CI: [0.90–0.99]). Manually segmented hematoma volumes and ABC/2 estimated volumes on pMRI correlate with conventional imaging volumes (ICC = 0.955, $p = 1.69e$-30 and ICC = 0.875, $p = 1.66e$-8, respectively). Hematoma volumes measured on pMRI correlate with NIH stroke scale (NIHSS) and clinical outcome (mRS) at discharge for manual and ABC/2 volumes. Low-field pMRI may be useful in bringing advanced MRI technology to resource-limited settings.

[1] Department of Neurology, Yale School of Medicine, New Haven, CT, USA. [2] Hyperfine Research, Inc, Guilford, CT, USA. [3] Department of Emergency Medicine, Yale School of Medicine, New Haven, CT, USA. [4] Department of Neurosurgery, Yale School of Medicine, New Haven, CT, USA. [5] Neuroscience Intensive Care Unit, Yale New Haven Hospital, New Haven, CT, USA. [6] Department of Radiology, Yale University School of Medicine, New Haven, CT, USA. [7] Athinoula A. Martinos Center for Biomedical Imaging, Massachusetts General Hospital, Charlestown, MA, USA. [8] Department of Neurology, Division of Neurocritical Care, Massachusetts General Hospital, Boston, MA, USA. [9]These authors contributed equally: Mercy H. Mazurek, Bradley A. Cahn. ✉email: wtkimberly@mgh.harvard.edu; kevin.sheth@yale.edu

Timely and accessible neuroimaging is a critical step in the diagnostic workup of patients presenting with suspected acute brain injury such as stroke[1,2]. Since intracerebral hemorrhage (ICH) is a contraindication for thrombolytic therapy[3,4], ruling out the presence of blood is one of the main decision steps in acute stroke care. Current guidelines for the early management of stroke from the American Heart Association (AHA) advise that all patients receive rapid brain imaging on hospital arrival prior to initiating any thrombolytic treatment[5]. Non-contrast computed tomography (CT) of the head has historically been the imaging modality of choice for diagnosing ICH due to its convenience and high sensitivity for hemorrhage[6–8]. However, a growing body of recent evidence has demonstrated that multimodal magnetic resonance imaging (MRI) is as accurate as CT for detecting acute brain hemorrhage[9–18] and avoids the radiation exposure associated with CT[19]. Certain strategies have previously been developed to reduce CT radiation burden[20,21]. Nevertheless, studies comparing CT and MRI demonstrate that magnetic resonance technology has higher sensitivity to ischemia, leukoencephalopathy, and classifying forms of extra-axial hemorrhage[22,23]. Furthermore, MRI is shown to offer more precise anatomic depiction of neuropathology and sharper resolution of soft tissue and contrast in comparison to CT[24,25].

In addition to acute stroke evaluation, other clinical contexts, such as post-neurosurgical assessment of patients, require neuroimaging evaluation to detect the presence of ICH. Neuroimaging is also essential for characterizing ICH, which aids in diagnosing the etiology of ICH, clinical management, and prognosis formation. For instance, non-lobar ICH is often caused by hypertension, and intraventricular and cerebellar hemorrhage require neurosurgical intervention for cerebrospinal diversion or suboccipital decompression. Additionally, clinicians commonly use ICH volume as a critical determinant of prognostication[26].

Traditionally, neuroimaging requires patient transport to a centralized dedicated radiology suite, which is costly in both time and resources[27–30]. Conventional MRI systems operate at high magnetic field strengths (1.5–3T)[31], which require specialized infrastructure, highly trained technicians, and rigid safety precautions[27,32,33]. As a result, MRI is not easily accessible for unstable patients or for populations in resource-limited settings where secure access radiology suites are not available throughout the day[34]. In ICH patients being transported for neuroimaging, potential adverse events include increased intracranial pressure, cardiovascular instability, and compromise of monitoring equipment and intravenous lines[35,36]. Recent advances in low-field MRI (<0.2T) have allowed for imaging outside of strict access-controlled radiology suites and in the presence of ferromagnetic materials at the point-of-care[27,37–41]. The ability to operate at low magnetic field strength eliminates the need for expensive superconducting magnets, can result in fewer susceptibility artifacts, and offers increased flexibility in open geometry design and improved $T_1$ contrast[27,38,40]. Previously, mid-field MRI technology (0.2–1T) has been employed for the acquisition of clinically useful imaging in critical care units[40,41] and the use of low-field MRI technology has been posited as a meaningful solution for stroke[25]. However, these efforts were based on large, fixed imaging systems rather than mobile, bedside units. A prior report presented a mobile and efficient low-field (23 mT) MRI prototype for neonatal applications, however no patient imaging was performed[37].

We report the use of a low-field (0.064T), portable MRI (pMRI) system (Fig. 1) in critically ill patients presenting with ICH. Our primary objective was to demonstrate the ability to deploy pMRI neuroimaging at the hospital bedside and provide initial evaluation for detection of ICH. We provide a systematic assessment of ICH detection using neuroimaging derived from pMRI. Specifically, we report the sensitivity and specificity of ICH detection and the accuracy of ICH localization. In addition, we explore the association between pMRI-derived ICH characteristics and clinical outcome.

## Results

**Study Cohort and Safety.** We obtained 119 pMRI examinations on 104 patients presenting to the neuroscience intensive care unit (NICU) or emergency department (ED) with a confirmed diagnosis of ICH or acute ischemic stroke (AIS). Eleven patients were imaged at serial timepoints. Nine exams were excluded from this analysis due to the patient having a body habitus that prevented full brain insertion into the scanner's head coil and produced an incomplete field-of-view. An additional six exams were significantly degraded due to patient motion and were excluded. The remaining 104 exams on 94 patients (42 women [45%]; median [IQR] age, 66 [19] years) were included in this analysis as ICH and AIS cases. Thirteen patients were imaged in the acute phase (≤24 h), 68 patients were imaged in the subacute phase (24 h to 7 days), and 13 patients were imaged in the chronic phase (>1 week). Patient NIHSS at the closest time to pMRI examination ranged from 0 to 37 (median 5) and functional outcome (mRS) at discharge ranged from 0 to 6 (median 3). Forty examinations of healthy controls (10 women [26%]; median [IQR] age, 50 [21] years) were included in this study. This included 24 healthy controls (5 women [21%]; median [IQR] age, 40 [14] years) scanned at Hyperfine HQ and 14 patients (5 women [36%]; median [IQR] age, 61 [11] years) with a diagnosis of no intracranial abnormality scanned at YNHH. Two healthy controls were serially imaged. Of the total cohort, 5 pMRI examinations reached the level of the midbrain (4%), 25 exams reached the level of the pons (17%), 107 exams reached the level of the medulla (74%), and 7 exams reached the level of the lateral ventricles (5%). Table 1 summarizes the demographic and clinical characteristics of the study population.

No adverse events occurred. Patients remained connected to all intravenous lines and ICU monitoring equipment during sequence acquisition. Table 2 delineates the time required for pMRI imaging compared to conventional MRI. The mean examination time was 17:51 min for a pMRI protocol that included pre-scan calibration, localizer, T2W, and FLAIR imaging. The static magnetic field, gradient, and RF pulses of the pMRI scanner did not interfere with the operation of infusion pumps, mechanical ventilators, or hemodialysis machines. Research associates and clinical staff were able to remain in the room during sequence acquisition.

**Sensitivity and specificity rater evaluation.** Neuroradiologists were provided a total of 144 exams (56 ICH, 48 AIS, 40 healthy controls) for evaluation. There were 52 cases of disagreement between the neuroradiologists that were evaluated by an ICH imaging core lab researcher. Exams were correctly classified as positive or negative for ICH in 130 of 144 total cases (90.3% overall accuracy; Gwet's AC2 = 0.791, 95% CI: [0.718–0.864], $p = 0$). ICH was detected in 45 of 56 cases (80.4% sensitivity, 95% CI: [0.68–0.90]; PPV = 0.94, 95% CI: [0.82–0.99]; NPV = 0.89, 95% CI: [0.80–0.94]). Ischemic stroke and healthy control cases were correctly identified as blood-negative in 85 of 88 cases (96.6% specificity, 95% CI: [0.90–0.99]). Primary ICH in a supratentorial location, which is the most common presentation of ICH[42,43], was correctly identified in 44 of 50 cases (88.0% sensitivity, 95% CI: [0.76–0.95]; PPV = 0.94, 95% CI: [0.82–0.99]; NPV = 0.93, 95% CI: [0.86–0.98]). Raters detected the presence of IVH in 13 of 14 ICH cases (92.8% sensitivity, 95% CI: [0.66–1.0];

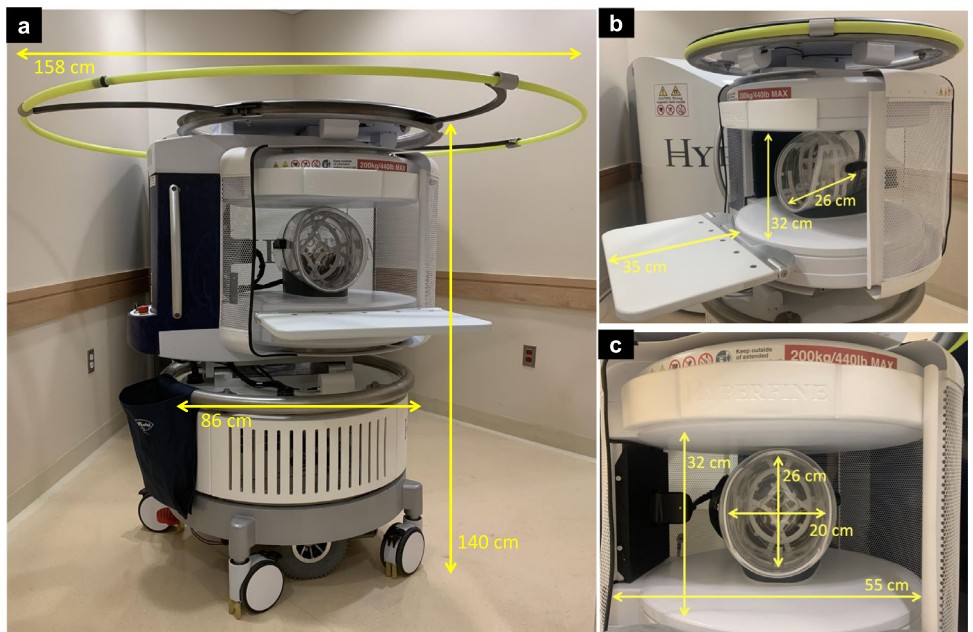

**Fig. 1 Portable (0.064T) magnetic resonance imaging device dimensions. a** The portable MRI (pMRI) device has a height of 140 cm and a width of 86 cm. The critical 5 Gauss (0.5 mT) boundary around the scanner extends into a circle with a diameter of 158 cm. **b** The pMRI device is positioned at the head of the patient's hospital bed. The scanner bridge (35 cm) adjoins the hospital bed with the pMRI device and the patient's chest height and head and neck lengths are positioned within the vertical clearance between magnets (32 cm) and the head coil length (26 cm), respectively. **c** The patient's head is positioned within the single channel transmit, 8-channel receiver head coil (26 × 20 cm) and the RF shield is closed for scan acquisition, which creates a horizontal clearance of 55 cm.

**Table 1 Demographic and clinical characteristics of study population.**

| Characteristics | All subjects (*n* = 132) | ICH (*n* = 50) | AIS (*n* = 44) | Healthy controls[a] (*n* = 38) |
|---|---|---|---|---|
| Age, mean (SD) | 60 (16) | 63 (16) | 65 (14) | 48 (14) |
| Sex, no. (%) | | | | |
| Male | 81 (61) | 26 (52) | 26 (59) | 28 (74) |
| Female | 51 (39) | 24 (48) | 18 (41) | 10 (26) |
| Race, no. (%) | | | | |
| White | 64 (68) | 32 (64) | 32 (73) | |
| Black/African American | 13 (14) | 8 (16) | 5 (11) | |
| Asian | 8 (9) | 5 (10) | 3 (7) | |
| Other | 6 (6) | 3 (6) | 3 (7) | |
| Unknown | 3 (3) | 2 (4) | 1 (2) | |
| Baseline medical history, no. (%) | | | | |
| Previous stroke | 12 (13) | 8 (16) | 4 (9) | |
| Hypertension | 58 (62) | 29 (58) | 29 (66) | |
| Hyperlipidemia | 36 (38) | 15 (30) | 21 (48) | |
| Diabetes mellitus | 21 (22) | 9 (18) | 12 (27) | |
| Atrial fibrillation | 14 (15) | 6 (12) | 8 (18) | |
| Time from LKN to exam, median (IQR), hrs | 55 (68) | 58 (84) | 55 (65) | |
| NIHSS at admission[b], median (IQR) | 5 (11) | 5 (13) | 6 (11) | |
| NIHSS at exam[c], median (IQR) | 5 (12) | 6 (11) | 6 (13) | |
| Functional outcome (mRS), median (IQR) | 4 (3) | 4 (3) | 4 (3) | |

*IQR* interquartile range, *LKN* last known normal, *NIHSS* National Institutes of Health Stroke Scale, *mRS* modified Rankin Scale.
[a]Race and baseline medical history were not collected for healthy control subjects.
[b]NIHSS at admission values were unavailable for 9 subjects.
[c]NIHSS at exam values were unavailable for 7 subjects.

PPV = 0.81, 95% CI: [0.54–1.0]; NPV = 0.99, 95% CI: [0.94–1.0]). Examples of true positive neuroimaging findings are shown in Fig. 2, compared to traditional high-field MRI or CT. Figure 3 demonstrates false negative neuroimaging findings for ICH.

To account for confounding effects due to evolving improvements in scanner software and hardware, an identical analysis to above was performed in two subsets by grouping exams by software versions into the first half of the study and its respective

scanner software versions (RC3, RC4, RC5), and the second half of the study and its respective scanner software versions (RC6, RC7, RC8).

Exams collected during the first half of scanner software versions were correctly classified in 54 of 60 cases (90.0% overall accuracy; Gwet's AC2 = 0.772, 95% CI: [0.651–0.892], *p* = 0). ICH was detected in 16 of 22 cases (72.7% sensitivity, 95% CI: [0.50–0.89]; PPV = 1.0, 95% CI: [0.79–0.96]; NPV = 0.86, 95%

**Table 2 Portable MRI examination time compared to conventional MRI.**

| Portable MRI[a] | | Conventional MRI[b] | |
|---|---|---|---|
| **Scan preparation** | **Time (mins:s)** | **Scan preparation** | **Time (mins:s)** |
| Prepare ICU room for pMRI scanner entry | 01:28 ± 0:02 | Prepare patient for transport | 05:56 ± 0:11 |
| Move scanner from hall to head of hospital bed | 00:49 ± 0:01 | Transport from ICU to holding room of radiology suite | 08:33 ± 0:12 |
| Position patient in pMRI scanner and initialize scan acquisition | 06:07 ± 0:09 | Prepare patient for entry into high field environment in holding room | 15:16 ± 0:43 |
| | | Transport from holding room, position in MRI gantry, and initialize scan acquisition | 05:05 ± 0:04 |
| **Sequence acquisition** | **Time (mins:s)** | **Sequence acquisition** | **Time (mins:s)** |
| Pre-scan calibration | 01:03 | Pre-scan calibration | 00:21 ± 0:01 |
| Localizer | 00:18 | Localizer | 00:19 ± 0:01 |
| T2W (axial) | 07:01 | T2W (axial) | 01:55 ± 0:01 |
| FLAIR (axial) | 09:29 | FLAIR (axial) | 02:47 ± 0:02 |
| **Scan termination** | **Time (mins:s)** | **Scan termination** | **Time (mins:s)** |
| Remove patient from pMRI scanner | 00:44 ± 0:01 | Remove patient from MRI gantry and transport to radiology holding room | 03:03 ± 0:03 |
| Remove scanner from ICU room | 00:34 ± 0:01 | Prepare patient for transport from radiology holding room to ICU | 13:14 ± 0:11 |
| Reset patient ICU room | 03:08 ± 0:02 | Transport patient from radiology suite holding room to ICU | 07:11 ± 0:04 |
| | | Reset patient ICU room | 04:21 ± 0:18 |
| Total Time: | 30:21 | Total time: | 67:36 |

*ICU* intensive care unit, *T2W* T2-weighted, *FLAIR* fluid-attenuated inversion recovery.
[a]Portable MRI scan preparation and termination times are averages (mean ± SD) recorded for five non-intubated patient scans acquired on the portable, 64mT MRI in the neuroscience intensive care unit (NICU) at Yale New Haven Hospital.
[b]Conventional MRI scan preparation and scan termination times are averages (mean ± SD) recorded for five non-intubated NICU patient scans acquired on a Siemens MAGNETOM Verio 3T eco and AVANTO 1.5T MRI scanner at Yale New Haven Hospital.

CI: [0.73–0.95]). Ischemic stroke and healthy control cases were correctly identified as blood-negative in 38 of 38 cases (100% specificity, 95% CI: [0.91–1.0]).

Exams collected during the second half of scanner software versions were correctly classified in 76 of 84 cases (90.5% overall accuracy; Gwet's AC2 = 0.805, 95% CI: [0.712–0.898], $p = 0$). ICH was identified in 29 of 34 cases (85.3% sensitivity, 95% CI: [0.69–0.95]; PPV = 0.91, 95% CI: [0.75–0.98]; NPV = 0.91, 95% CI: [0.80–0.97]). Ischemic stroke and healthy control cases were correctly identified as blood-negative in 47 of 50 cases (94.3% specificity, 95% CI: [0.84–0.99]).

**Appearance of intracerebral hemorrhage on POC MRI.** ICH lesions were found in 50 supratentorial (89.3%) and 6 infratentorial (10.7%) locations. Supratentorial locations included the frontal lobe ($n = 8$), parietal lobe ($n = 2$), occipital lobe ($n = 5$), temporal lobe ($n = 5$), fronto-parietal lobes ($n = 9$), fronto-temporal lobes ($n = 2$), parieto-occipital lobes ($n = 2$), parieto-temporal lobes ($n = 2$), basal ganglia ($n = 7$), and thalamus ($n = 7$). One lesion appeared at the corpus callosum. Infratentorial locations included the pons ($n = 1$) and cerebellum ($n = 5$). Fourteen lesions (25.0%) had intraventricular components. Twenty-eight lesions (50.0%) were left-sided, 22 lesions (39.3%) were right-sided, and 6 patients (10.7%) had bilateral lesions.

Twenty-seven lesions on T2W appeared hypointense (SIR = 0.81 ± 0.10) with a hyperintense rim (1.27 ± 0.08), thirteen lesions appeared hyperintense (1.24 ± 0.10) with a hyperintense rim (1.73 ± 0.16), seven lesions appeared isointense (1.00 ± 0.08) with a hyperintense rim (1.41 ± 0.09), and nine lesions appeared as a homogenous hyperintensity (1.59 ± 0.11).

Fifteen lesions on FLAIR exams appeared hypointense (0.84 ± 0.03) with a hyperintense rim (1.24 ± 0.04), 8 lesions appeared hyperintense (1.20 ± 0.06) with a hyperintense rim (1.98 ± 0.10), 9 lesions appeared isointense (1.00 ± 0.05) with a hyperintense rim (1.31 ± 0.04), and 22 lesions appeared as a homogenous hyperintensity (1.60 ± 0.09). Two FLAIR pMRI cases on the earliest software version were excluded since the lesion was not visualized.

**Hematoma volume and localization.** Low-field pMRI examinations were obtained for 56 ICH patients. Fourteen exams were excluded from this sub-analysis since they did not have a conventional imaging study (CT or 1.5/3T MRI) within 36 h of the pMRI examination. As a result, 42 ICH patient exams were provided to one rater for manual segmentation. Five T2W and two FLAIR pMRI cases were excluded since their corresponding conventional sequence was not obtained; an additional two FLAIR pMRI cases on the earliest software version were excluded since the lesion was not visualized to segment. Manually segmented lesions on pMRI T2W and FLAIR sequences had a median [IQR] of 5.58 cc [3.38–16.9] and 8.61 cc [3.25–28.4], respectively. To estimate hematoma volume using the ABC/2 method[44], 40 of the 42 ICH patient exams were provided to four raters for evaluation; two lesions with non-ellipsoid morphology were excluded due to the inaccuracy of the ABC/2 measurement in these cases[45] and two FLAIR pMRI on cases the earliest software version were excluded since the lesion was not visualized. Hematoma volumes estimated using the ABC/2 method on pMRI T2W and FLAIR sequences were averaged across raters and had a median [IQR] of 7.93 cc [3.36–23.9] and 7.12 cc [2.67–22.1], respectively. There was significant interrater agreement on pMRI (ICC = 0.968, 95% CI: [0.953–0.978], $p = 1.41e\text{-}74$) and conventional (ICC = 0.978, 95% CI: [0.966–0.987], $p = 1.41e\text{-}61$) hematoma volume measurements using the ABC/2 method.

Low-field pMRI volumes were validated against the closest conventional exam (CT or 1.5/3T MRI) within 36 h. Hematoma volumes manually segmented on pMRI strongly correlated with conventional imaging volumes (T2W: ICC = 0.971, 95% CI: [0.940–0.985], $p = 4.99e\text{-}17$; FLAIR: ICC = 0.947, 95% CI: [0.898–0.972], $p = 2.69e\text{-}15$; T2W and FLAIR: ICC = 0.955, 95% CI: [0.928–0.971], $p = 1.69e\text{-}30$) (Fig. 4a1). Bland–Altman

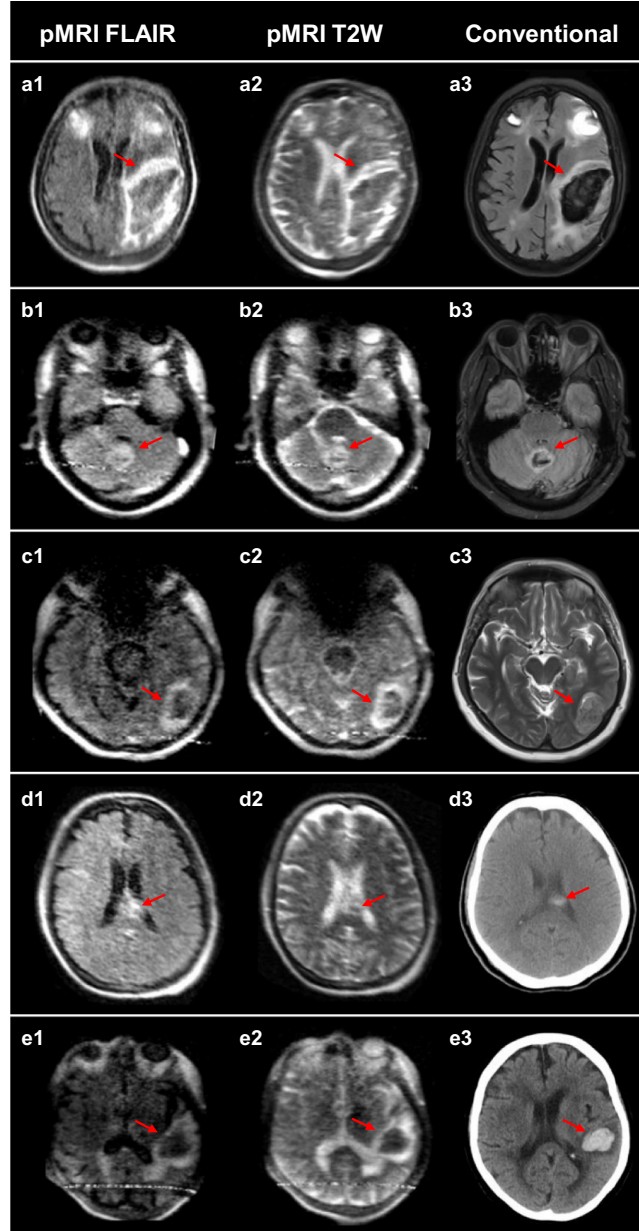

**Fig. 2 Intracerebral hemorrhage at 0.064T versus conventional imaging modalities (CT or 3T MRI).** The first and second columns are low-field FLAIR and T2W images, respectively. The third column is a gold-standard clinical examination for comparison (3T MRI: **a**3, **b**3, **c**3, and **e**3; CT: **d**3). **a** Left isointense fronto-parietal intracerebral hemorrhage (ICH) with hyperintense rim and bilateral frontal hematomas. **b** Bilateral isointense cerebellar ICH with hyperintense rim. **c** Left hypointense occipital lobe ICH with hyperintense rim. **d** Left homogenous, hyperintense ICH in corpus collosum. **e** Left hypointense temporal ICH with hyperintense rim.

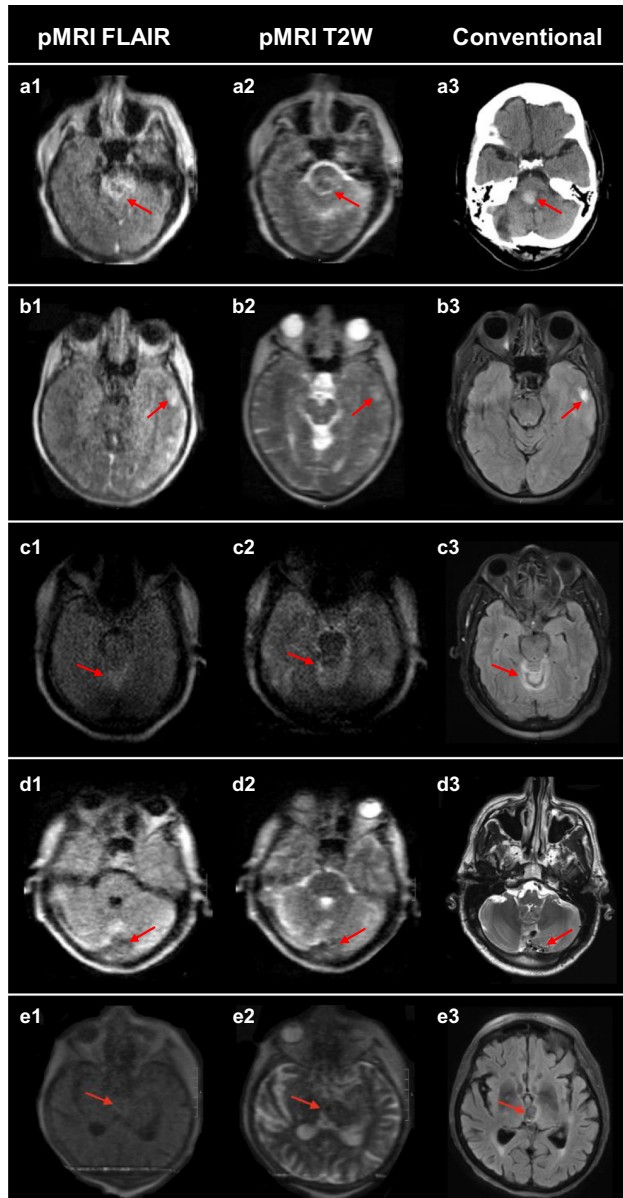

**Fig. 3 False negative intracerebral hemorrhage cases.** The first and second columns are low-field FLAIR and T2W images, respectively. The third column is a gold-standard clinical examination for comparison. (3T MRI: b3, d3, and e3; CT: a3). **a** Right cerebellar pontine intracerebral hemorrhage (ICH). Missed by all raters. **b** Left temporal ICH. Missed by all raters. **c** Bilateral cerebellar ICH. Missed by 2/3 raters. **d** Left cerebellum ICH. Missed by all raters. **e** Left thalamus ICH. Missed by all raters.

plots showed a bias of −1.70 cc [95% CI: −3.45–0.0470] with limits of agreement from −11.8 cc [95% CI: −14.8–8.81] to 8.42 cc [95% CI: 5.409–11.4] for pMRI T2W sequences (Fig. 4a2) and a bias of −1.22 cc [95% CI: −4.51–2.07] with limits of agreement from −20.8 cc [95% CI: −26.5 to −15.2] to 18.4 cc [95% CI: 12.7–24.1] for pMRI FLAIR sequences (Fig. 4a3).

Hematoma volumes estimated by the ABC/2 method were averaged across raters and strongly correlated with averaged conventional imaging volumes (T2W: ICC = 0.917, 95% CI: [0.814–0.96], $p = 1.58e-8$; FLAIR: ICC = 0.857, 95% CI: [0.665–0.932], $p = 9.46e-6$; T2W and FLAIR: ICC = 0.875, 95%

CI: [0.754–0.931], $p = 1.66e-8$) (Fig. 4b1). Bland–Altman plots showed a bias of −3.74 cc (95% CI: −6.62 to −0.855] with limits of agreement from −20.2 cc [95% CI: −25.2 to −15.2] to 12.7 cc [95% CI: 7.74 to 17.7] for pMRI T2W sequences (Fig. 4b2) and a bias of −7.89 cc [95% CI: −12.9 to −2.85] with limits of agreement from −38.0 cc [95% CI: −46.6 to −29.3] to 22.2 cc [95% CI: 13.5–30.9] for pMRI FLAIR sequences (Fig. 4b3).

Manually segmented hematoma volumes strongly correlated with estimated hematoma volumes measured by the ABC/2 method (T2W: ICC = 0.975, 95% CI: [0.937–0.988], $p = 1.71e-11$; FLAIR: ICC = 0.985, 95% CI: [0.959–0.994], $p = 4.89e-11$; T2W and FLAIR: ICC = 0.98174, 95% CI: [0.957–0.991], $p = 2.62e-14$) (Fig. 4c1). Bland–Altman plots showed a bias of 1.96 cc [95% CI: 0.609–3.32] with limits of agreement from −5.76 cc [95% CI:

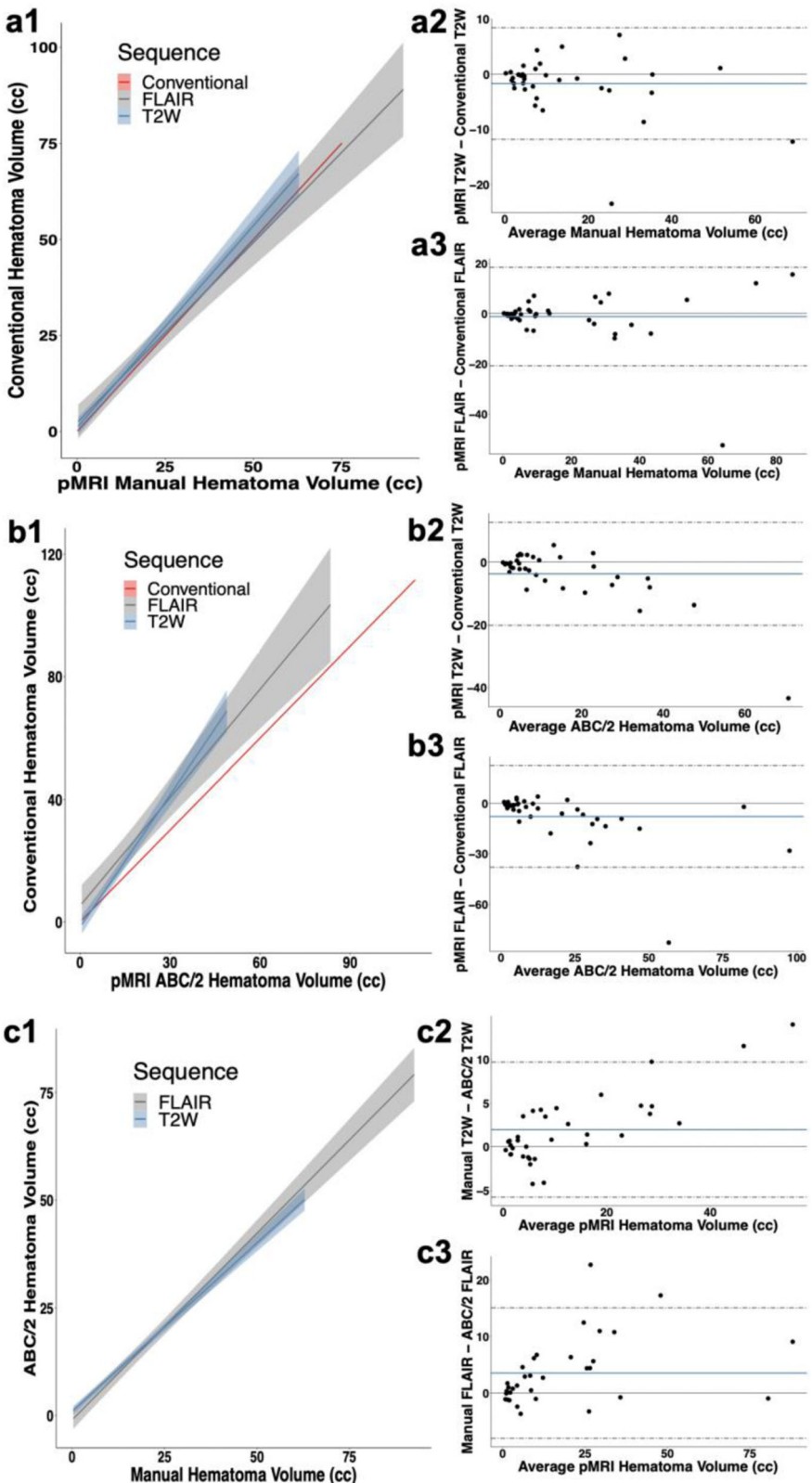

−8.09 to −3.42] to 9.68 cc [95% CI: 7.35–12.0] for pMRI T2W sequences (Fig. 4c2) and a bias of 1.79 cc [95% CI: 0.695–2.89] with limits of agreement from −4.46 cc [95% CI: −6.35 to −2.57] to 8.04 cc [95% CI: 6.15–9.94] for pMRI FLAIR sequences (Fig. 4c3).

To localize lesions between pMRI and conventional (CT or 1.5/3T MRI) examinations, the Euclidean distances between the centroids of

the manually segmented hemorrhages and the centroids of each point ROI at the aforementioned anatomical locations were compared across modalities. There was significant correlation between Euclidean distances to the optic chiasm (T2W: ICC = 0.962, 95% CI: [0.923–0.981], $p$ = 1.32e-15; FLAIR: ICC = 0.951, 95% CI: [0.906–0.974], $p$ = 4.61e-16), the septum pellucidum (T2W: ICC = 0.918, 95% CI: [0.834–0.959], $p$ = 1.03e-10; FLAIR: ICC = 0.977,

**Fig. 4 Hematoma volume measurements on portable MRI. a1** Validation of manually segmented pMRI hematoma volumes against manual volumes on conventional (CT or 1.5/3T MRI) imaging (T2W ($n = 37$): $r = 0.952$, 95% CI: [0.907-0.975], $p < 2.20\text{e-16}$; FLAIR ($n = 38$): $r = 0.899$, 95% CI: [0.812-0.946], $p = 1.90\text{e-14}$). Bland-Altman plots for manual pMRI showed a bias of $-1.70$ cc [limits of agreement (LOA): $-11.8$-8.42] for (**a2**) T2W sequences ($n = 37$) and a bias of $-1.22$ cc [LOA: $-20.8$-18.4] for (**a3**) FLAIR ($n = 38$). **b1** Validation of averaged ABC/2 estimated pMRI volumes against averaged estimated volumes on conventional (CT or 1.5/3T MRI) imaging (T2W ($n = 40$): $r = 0.945$, 95% CI: [0.892-0.972], $p < 2.20\text{e-16}$; FLAIR ($n = 38$): $r = 0.835$, 95% CI: [0.702-0.911], $p = 7.53\text{e-11}$). Bland-Altman plots for ABC/2 pMRI showed a bias of $-3.74$ cc [LOA: $-20.2$-12.7] for (**b2**) T2W ($n = 40$) and a bias of $-7.89$ [LOA: $-38.0$-22.2] for (**b3**) FLAIR ($n = 38$). **c1** Manually segmented pMRI hematoma volumes against averaged estimated volumes using ABC/2 (T2W ($n = 37$): $r = 0.977$, 95% CI: [0.956-0.989], $p < 2.20\text{e-16}$; FLAIR ($n = 38$): $r = 0.968$, 95% CI: [0.936-0.984], $p < 2.20\text{e-16}$). Bland-Altman plots showed a bias of 1.962 [LOA: $-5.76$-9.68] for (**c2**) T2W ($n = 37$) and a bias of 1.79 [LOA: $-4.46$-8.04] for (**c3**) FLAIR ($n = 38$). Pearson correlations are reported for **a**1, **b**1, and **c**1 with confidence intervals. Line of identity shown in red (**a**1, **b**1). 95% confidence intervals are represented by bands (**a**1, **b**1, **c**1) and dashed gray lines (**a**2-3, **b**2-3, **c**2-3).

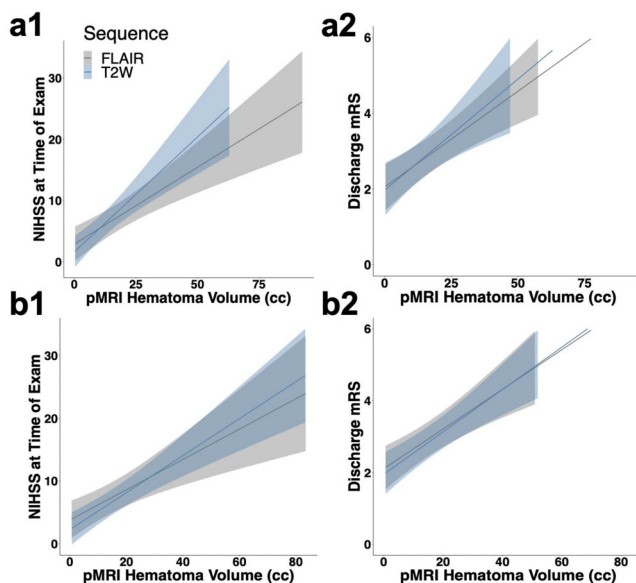

**Fig. 5 Hematoma volume and cognitive scores on portable MRI.** Manual pMRI hematoma volume versus (**a**1) cognitive status (NIHSS) at time of exam (pMRI T2W ($n = 33$): $\rho = 0.750$, 95% CI: [0.591-0.906], $p = 4.95\text{e-}7$; pMRI FLAIR ($n = 34$): $\rho = 0.802$, 95% CI: [0.669-0.930], $p = 1.23\text{e-8}$) and **a**2 functional status (mRS) at discharge (pMRI T2W ($n = 36$): $\rho = 0.589$, 95% CI: [0.372-0.804], $p = 1.55\text{e-4}$; pMRI FLAIR ($n = 37$): $\rho = 0.641$, 95% CI: [0.425-0.855], $p = 1.89\text{e-5}$). Averaged ABC/2 estimated pMRI hematoma volume versus (**b**1) cognitive status (NIHSS) at time of exam (pMRI T2W ($n = 36$): $\rho = 0.805$, 95% CI: [0.659-0.948], $p = 3.18\text{e-9}$; pMRI FLAIR ($n = 34$): $\rho = 0.776$, 95% CI: [0.617-0.930], $p = 7.16\text{e-8}$) and (**b**2) functional status (mRS) at discharge (pMRI T2W ($n = 39$): $\rho = 0.747$, 95% CI: [0.592-0.899], $p = 4.85\text{e-8}$; pMRI FLAIR ($n = 37$): $\rho = 0.669$, 95% CI: [0.483-0.853], $p = 5.98\text{e-6}$). Spearman correlations are reported for **a**1, **a**2, **b**1, and **b**2 with confidence intervals. Bands represent 95% confidence intervals.

95% CI: [0.955-0.988], $p = 3.12\text{e-21}$), the right edge of the central sulcus (T2W: ICC $= 0.930$, 95% CI: [0.861-0.965], $p = 3.44\text{e-12}$; FLAIR: ICC $= 0.945$, 95% CI: [0.895-0.971], $p = 5.18\text{e-15}$), and the left edge (T2W: ICC $= 0.982$, 95% CI: [0.964-0.991], $p = 2.27\text{e-21}$; FLAIR: ICC $= 0.984$, 95% CI: [0.970-0.992], $p = 2.37\text{e-24}$). Pearson correlations and Bland-Altman statistics are summarized in Table S1 and plots shown in Fig. S1.

**Hematoma volume and clinical outcome.** Hematoma volume measurements derived from pMRI exams were associated with impaired cognitive status and worse discharge functional outcome as assessed by the National Institutes of Health Stroke Scale (NIHSS) and the Modified Rankin Scale (mRS), respectively. Hematoma volume manually segmented on pMRI exams

significantly correlated with cognitive status (NIHSS) at time of exam (T2W: $\rho = 0.750$, 95% CI:[0.591-0.906], $p = 4.95\text{e-7}$; FLAIR: $\rho = 0.802$, 95% CI:[0.669-0.930], $p = 1.23\text{e-8}$) (Fig. 5a1) and functional outcome (mRS) at discharge (T2W: $\rho = 0.589$, 95% CI:[0.372-0.804], $p = 1.55\text{e-4}$; FLAIR: $\rho = 0.641$, 95% CI:[0.425-0.855], $p = 1.89\text{e-5}$) (Fig. 5a2). For T2W sequences, hematoma volume measurements predicted cognitive status at time of exam (NIHSS) (unadjusted coefficient 0.376, 95% CI [0.228-0.523], $p = 1.27\text{e-5}$; adjusted coefficient 0.375, 95% CI [0.209-0.541], $p = 8.52\text{e-5}$) and discharge functional outcome (mRS) (unadjusted coefficient 0.0587, 95% CI [0.0247-0.0928], $p = 1.31\text{e-3}$; adjusted coefficient 0.0629, 95% CI [0.0278-0.0980], $p = 1.13\text{e-2}$). For FLAIR sequences, hematoma volume measurements predicted cognitive status at time of exam (NIHSS) (unadjusted coefficient 0.2510, 95% CI [0.1460-0.3561], $p = 2.93\text{e-5}$; adjusted coefficient 0.269, 95% CI [0.157-0.381], $p = 4.16\text{e-5}$) and discharge functional outcome (mRS) (unadjusted coefficient 0.0506, 95% CI [0.0282-0.0731], $p = 5.68\text{e-5}$; adjusted coefficient 0.0552, 95% CI [0.0306-0.0797], $p = 7.54\text{e-5}$) (Table 3).

Hematoma volumes estimated by the ABC/2 method on pMRI exams and averaged across raters significantly correlated with cognitive status (NIHSS) at time of exam (T2W: $\rho = 0.805$, 95% CI:[0.659-0.948], $p = 3.18\text{e-9}$; FLAIR: $\rho = 0.776$, 95% CI:[0.617-0.930], $p = 7.16\text{e-8}$) (Fig. 5b1) and functional outcome (mRS) at discharge (T2W: $\rho = 0.747$, 95% CI:[0.592-0.899], $p = 4.85\text{e-8}$; FLAIR: $\rho = 0.669$, 95% CI:[0.483-0.853], $p = 5.98\text{e-6}$) (Fig. 5b2). For T2W sequences, averaged hematoma volume measurements predicted cognitive status at time of exam (NIHSS) (unadjusted coefficient 0.376, 95% CI [0.228-0.523], $p = 1.27\text{e-5}$; adjusted coefficient 0.375, 95% CI [0.209-0.541], $p = 2.55\text{e-3}$) and discharge functional outcome (mRS) (unadjusted coefficient 0.0587, 95% CI [0.0247-0.0928], $p = 1.31\text{e-3}$; adjusted coefficient 0.0623, 95% CI [0.0278-0.0980], $p = 1.13\text{e-2}$). For FLAIR sequences, averaged hematoma volume measurements predicted cognitive status at time of exam (NIHSS) (unadjusted coefficient 0.251, 95% CI [0.146-0.356], $p = 2.93\text{e-5}$; adjusted coefficient 0.269, 95% CI [0.157-0.381], $p = 6.33\text{e-4}$) and discharge functional outcome (mRS) (unadjusted coefficient 0.0506, 95% CI [0.0282-0.0731], $p = 5.68\text{e-5}$; adjusted coefficient 0.0552, 95% CI [0.0306-0.0787], $p = 1.58\text{e-3}$) (Table 3).

## Discussion

We report the validation of portable MRI (pMRI) in evaluating intracerebral hemorrhage (ICH). These results demonstrate the successful deployment of a low-field pMRI device to the bedside of critically ill patients with ICH. With this approach, we obtained neuroimaging results that enabled detection and characterization of ICH. Our observation that ICH volume measured on pMRI is associated with both stroke severity and patient outcome further validates this approach as ICH volume is a well-established predictor of outcome[46]. These results suggest that

**Table 3 Portable MRI hematoma volume and patient cognitive scores.**

| pMRI sequence | Method | Cognitive scores | Unadjusted | | Adjusted | |
|---|---|---|---|---|---|---|
| | | | Coefficient (95% CI) | P Value | Coefficient (95% CI) | P Value |
| T2W | Manual | NIHSS | 0.376 (0.226–0.523) | 1.27e-5 | 0.375 (0.209–0.541) | 8.52e-5 |
| T2W | Manual | Discharge mRS | 0.0587 (0.0247–0.0928) | 1.31e-3 | 0.0629 (0.0278–0.0980) | 1.13e-2 |
| FLAIR | Manual | NIHSS | 0.251 (0.146–0.356) | 2.93e-5 | 0.269 (0.157–0.381) | 4.16e-5 |
| FLAIR | Manual | Discharge mRS | 0.0506 (0.0282–0.0731) | 5.68e-5 | 0.0552 (0.0306–0.0797) | 7.54e-5 |
| T2W | ABC/2 | NIHSS | 0.376 (0.228–0.523) | 1.27e-5 | 0.375 (0.209–0.541) | 2.55e-3 |
| T2W | ABC/2 | Discharge mRS | 0.0587 (0.0247–0.0928) | 1.31e-3 | 0.0629 (0.0278–0.0980) | 1.13e-2 |
| FLAIR | ABC/2 | NIHSS | 0.251 (0.146–0.356) | 2.93e-5 | 0.269 (0.157–0.381) | 6.33e-4 |
| FLAIR | ABC/2 | Discharge mRS | 0.0506 (0.0282–0.0731) | 5.68e-5 | 0.0552 (0.0306–0.0787) | 1.58e-3 |

Hematoma volume measurements derived from pMRI exams were associated with impaired cognitive status and worse discharge functional outcome in unadjusted and adjusted multivariable linear regression models. Adjusted linear regression models included sex, race, and age.
pMRI portable magnetic resonance imaging, T2W T2-weighted, FLAIR fluid-attenuated inversion recovery, NIHSS National Institutes of Health Stroke Scale, mRS Modified Rankin Scale.

pMRI-based neuroimaging assessments are a point-of-care solution that could be useful in a broad range of clinical settings for diagnosis and evaluation.

Neuroimaging is the most reliable method of differentiating ICH from non-ICH etiologies of focal neurological deficit. Observational series two decades earlier provided some of the first evidence demonstrating at least equivalent sensitivity for MRI in detecting ICH compared to CT[9]. As many as 70% of patients undergo MRI during a hospital admission. However, the timing and availability of MRI evaluation is routinely limited by patient transport, access to secure access facilities, and safety concerns, either due to an individual's clinical status or potential danger in a high magnetic field strength environment. The risks inherent in intrahospital transport to dedicated radiology suites (CT or 1.5/3T MRI) are reported between 20–70%, even under the supervision of a well-trained transport team[47,48]. While portable CT scanners have been introduced to address these concerns[29,30,35,47,49–52], the lower spatial resolution, amplified noise, and higher radiation dose compared to their fixed-location counterpart render them disadvantageous to broad adoption in a dynamic hospital setting[47,50,52]. Moreover, the portable CT scanners require trained technicians and lead shielding to operate[49]. Conversely, the low-field pMRI solution can be used in the absence of a specialized MRI technician and requires minimal training of operator staff. Low-field pMRI does not have a liquid cryogenic requirement or use ionizing radiation, permits unrestricted access into the hospital room during acquisition, and allows for the added benefits of MRI evaluation compared to CT[22–25].

Using a pMRI approach, we demonstrate that ICH can be detected at the bedside using low-field (0.064T) magnetic resonance technology. This approach allows for a reversal in the clinical paradigm, wherein the pMRI comes to the patient. Our results extend our preliminary success in deploying a pMRI solution to the bedside of critically ill patient populations[53]. Due to the low magnetic field strength of the device, hospital staff and patients were able to safely enter the clinical environment, with no need to remove ferromagnetic objects required for clinical care. The scanner's open geometry design allowed for easy access to intravenous lines, ventilation tubing, and intraventricular drains during pMRI examinations. Patients and research staff did not experience any adverse events during pMRI deployment and could safely remain in the hospital room during scan acquisition.

Our objective was to characterize the presence of ICH in a brain-injured population focusing on characteristics that are known to have clinical relevance, including detection of ICH, description of anatomical location, and hematoma volume. The overall sensitivity for ICH was notable, especially for supratentorial ICH, which is the most common presentation[42,43]. However, because detection and accurate measurement of ICH is essential for acute stroke care and

hemorrhage in the posterior fossa is the more life-threatening subtype[54], further improvements in pMRI hardware and software will be required. Indeed, sensitivity improvements were observed even during this study as upgraded hardware and software versions became available. Furthermore, our results demonstrate that hematoma volumes measured on pMRI are in agreement with those measured on conventional imaging. To further validate this volumetric approach, the Euclidean distances between the centroids of the manually segmented hematomas and the centroids of the point ROIs at four anatomical locations were compared across modalities. We observed a strong correlation between distances across modalities, demonstrating congruence of lesion localization on pMRI with conventional imaging. Overall, further study is required to determine how detection can be most efficiently improved with hardware and software advances, image reconstruction tools, automated detection strategies, and reader training.

Exams from six patients were not interpretable due to motion artifacts and degraded images. The low field strength of the magnet and the resulting lower signal-to-noise ratio makes these images particularly susceptible to patient motion. Domain transform manifold learning is one of several post-processing techniques in the literature that we speculate could be applied to pMRI reconstruction to mitigate this limitation in the future[55]. Future clinical utility will require further quality improvements in motion-related artifact. Furthermore, exams from nine patients were excluded from this analysis due to the patient having a body habitus that prevented full entry into the scanner's opening and produced a limited field-of-view. While this small subset of patients obtained a pMRI exam with a field-of-view that did not capture the infratentorial pathology, the majority of patients had a complete pMRI field-of-view that reached to the level of the medulla and pons, allowing for full brain insertion and infratentorial evaluation.

It is also important to note the patient populations that our study did not fully evaluate. Low-field pMRI examination largely occurred within the subacute phase and future studies will need to evaluate ICH in the hyperacute setting, given differences in ICH appearance on MRI at different time points[56,57]. In addition, our study did not evaluate for the presence of subarachnoid hemorrhage. Along with improved detection of ICH in infratentorial compartments and evaluation of subarachnoid hemorrhage, the results in the current study should be confirmed in larger multicenter studies that investigate all forms of intracranial hemorrhage. In the future, it would be beneficial to conduct a systematic study to help us understand which populations can and cannot receive pMRI scans across a variety of applications.

There are several strengths in the current analysis. First, we were able to integrate into the clinical workflow an innovative

solution that facilitated bedside detection of ICH using MRI technology. Second, we used several blinded raters, all with established expertize reading clinical neuroimaging studies but with a range of experience, supporting generalizability of the clinical interpretation. In using a large, systematic dataset we are able to report both the proof-of-concept as well as the limitations of detection in a single-center cohort, acknowledging that point estimates for detection will require further validation in large cohorts across multiple care environments. Moreover, demonstrating an established relationship between pMRI-ascertained ICH volume with cognitive impairment and patient outcome provides additional support of pMRI as a bedside neuroimaging solution. We also report initial signal intensity characteristics of the lesions seen on T2W and FLAIR sequences using pMRI, which have been used to interrogate mechanisms of injury. Lastly, we were able to obtain serial imaging with the pMRI in this study, suggesting that pMRI may be used to understand dynamic changes in ICH pathology over time, which is a structural limitation in current MRI approaches. These results demonstrate that pMRI is a safe and feasible neuroimaging solution that contributes valuable information in ICH evaluation. Overall, we report a systematic assessment of ICH using a portable MRI device that can be integrated into the clinical workflow to provide timely diagnostic neuroimaging at the bedside. Further study is required in prospective multicenter studies.

## Methods

**Setting, participants, and study design**. This observational study was performed at Yale New Haven Hospital (YNHH) in New Haven, Connecticut from July 2018 to November 2020 under an Institutional Review Board (IRB) protocol approved by Yale Human Research Protection Program. From July 2018 to March 2020, the study was conducted with an investigational device exemption (IDE) and informed consent was obtained from all patients or their legally authorized representative. Starting in March 2020, pMRI neuroimaging examinations were obtained as part of the patients' clinical care under U.S. Food and Drug Administration (FDA) general clearance for portable imaging systems during the COVID-19 public health emergency[58]. The pMRI scanner later received specific FDA approval on 11 August 2020, and pMRI examinations continued as part of the patients' clinical care. This study complied to all relevant ethical regulations as outlined by the Yale Human Research Protection Program and U.S. FDA. Individuals who presented with a focal neurological deficit and were admitted to the emergency department (ED) or neuroscience intensive care unit (NICU) were screened for eligibility. Participant eligibility was determined based on admitting diagnosis, clinical exam, and baseline standard-of-care imaging during hospitalization. Bedside pMRI examinations were not obtained on patients with the presence of at least one of the following contraindications to conventional MRI evaluation: cardiac pacemakers or defibrillators, implantable drug pumps, deep brain stimulators, vagus nerve stimulators or cochlear implants, pregnancy, and cardiorespiratory instability.

Patients who received a pMRI examination with the following inclusion criteria were included in this analysis as stroke patients[1]: a clinical diagnosis of intracerebral hemorrhage (ICH) or acute ischemic stroke (AIS) confirmed on the radiological report of the closest conventional imaging study (1.5/3T MRI or CT)[2]; the acquisition of both T2-weighted (T2W) and Fluid-Attenuated Inversion Recovery (FLAIR) pMRI sequences; and[3] a patient body habitus that permitted positioning for full brain imaging within the scanner's head coil. Patients who received a pMRI examination with the following inclusion criteria were included in this analysis as healthy controls[1]: no past medical history of neuropathology;[2] a confirmed diagnosis of no intracranial abnormality on all available standard-of-care imaging reports;[3] both T2W and FLAIR pMRI sequences obtained; and[4] a patient body habitus that permitted positioning for full brain imaging within the scanner's head coil. Demographic characteristics, medical history, baseline functional independence, clinical presentation, discharge functional outcome, and standard-of-care hospital imaging findings were collected for each subject from the electronic medical record.

In addition, non-patient healthy controls were recruited at Hyperfine headquarters (HQ) in Guilford, CT, USA. All healthy control imaging at Hyperfine HQ was performed under a protocol approved by the Western Institutional Review Board-Copernicus Group (WCG) local IRB and written informed consent was obtained prior to imaging. Each exam consisted of two-dimensional multi-slice, whole brain axial T2W and FLAIR sequence imaging. The authors affirm that all patient participants enrolled from July 2018 to March 2020 and non-patient human research participants enrolled at Hyperfine HQ provided informed consent for publication of the data in Table 1 and the images presented in Figs. 2 and 3. Patients receiving pMRI examinations as part of clinical care from March 2020 to November

2020 did not provide informed consent as the requirement for informed consent to publish the data was waived by the Yale Human Research Protection Program.

**Portable MRI specifications**. All patients and healthy controls were examined with a bedside pMRI system (Hyperfine Research, Inc., Guilford, CT, USA) at 0.064T static magnetic field strength. The pMRI device has a height of 140 cm, a width of 86 cm, and weighs 630 kg. The critical 5 Gauss (0.5 mT) boundary around the scanner extends into a circle with a diameter of 158 cm and is deployed when the device is in transit. The scanner is positioned at the head of the patient's hospital bed. The scanner bridge adjoins the hospital bed with the pMRI device and has a length of 35 cm. The vertical clearance between magnets and horizontal clearance between closed RF shields are 32 cm and 55 cm, respectively. The head coil has a length of 26 cm, a height of 26 cm, and a width of 20 cm (Fig. 1). The pMRI device can be transported by elevator and requires a doorway clearance >86 cm wide. Power was drawn from a standard 15A, 110 V wall outlet in the patient's hospital room or in the mock ICU at Hyperfine HQ. The decibel level inside the coil ranges from 60 to 80 db. The pMRI uses a biplanar, whole body (unshielded) gradient system which includes X, Y, and Z direction gradient coils, enabling imaging in any 3D orientation similar to conventional fixed-location MR scanners. The peak amplitudes of the gradients are 25 mT/m on the $X$- and $Y$-axis and 26 mT/m on the $Z$-axis. The gradient peak slew rates are 23 T/m/s on the $X$- and $Y$-axis and 67 T/m/s on the $Z$-axis. Images were acquired using a single channel transmit, 8-channel receiver head coil.

The scanning environment contained ferrous metal and standard intensive care unit equipment, including but not limited to the electrocardiogram and vital signs monitor, IV infusion pumps, ventilators, compressed gas tanks, and dialysis machines. Exams were administered by research staff trained to operate the pMRI scanner without the need for a specialized MR technician. Scan sequences were controlled using a tablet computer interface (iPad Pro, 2nd generation and 3rd generation; Apple, Cupertino, CA, USA).

The pMRI device underwent multiple hardware and software versions over the course of the study (hardware Mk1.2, Mk1.5, Mk1.6; software versions RC3, RC4, RC5, RC6, RC7, RC8). Pulse sequences were pre-configured by the manufacturer for all patients. All exams were acquired in the axial plane. For T2W fast spin echo (FSE) imaging, relevant acquisition parameters were organized as follows (RC8/RC5/RC3): echo time [TE] = 252.6/252.3/200.5 ms, repetition time [TR] = 2200/2000/2000 ms, echo train length = 80/72/64, number of averages = 1/1/1, resolution = $1.5 \times 1.5 \times 5$ mm$^3$/$1.5 \times 1.5 \times 5$ mm$^3$/$1.7 \times 1.7 \times 5$ mm$^3$, slices = 36/36/36, acquisition time: 7:01/5:28/8:39 min. For FLAIR FSE, relevant acquisition parameters were: [TE] = 227.5/172.6/155.28 ms, [TR] = 4000/100/1000 ms, inversion time [TI] = 2500/350/350, echo train length = 80/48/48, number of averages = 1/1/1, resolution: $1.6 \times 1.6 \times 5$ mm$^3$/$1.5 \times 1.5 \times 5$ mm$^3$/$1.7 \times 1.7 \times 5$ mm$^3$, slices = 36/36/36, acquisition time = 9:29/8:11/8:35 min.

The scanner comes with a system quality assurance phantom, which is scanned on a monthly basis for calibration and monitoring purposes. The phantom image sets are uploaded to the Hyperfine Cloud Picture Archive and Communication System (PACS) and evaluated by Hyperfine Research, Inc. clinical scientists. In addition, each scanner passes a phantom-based factory acceptance test before delivery, which verifies performance metrics established according to National Electrical Manufacturers Association (NEMA) standards, and the results are reported to the U.S. FDA.

**Sensitivity and specificity rater evaluation**. A panel of two board-certified neuroradiologists (G.S. 40 years of experience, A.M. 22 years of experience) independently evaluated T2W and FLAIR pMRI exams. An ICH imaging core lab researcher (A.L.) with 7 years of experience reviewing clinical neuroimaging of ICH adjudicated the cases of disagreement between neuroradiologists.

Readers were blinded to all clinical and demographic information. The order of patients was randomized and both pulse sequences (T2W and FLAIR) were interpreted simultaneously in Horos (v3.3.5) or RadiAnt (v.2020.2.3). T2W and FLAIR imaging sequences were selected for rater evaluation since these were the only two sequences that were consistently available on the device across software changes occurring from the beginning of the study (July 2018) to its completion (November 2020).

For each pMRI exam, readers recorded the following data: presence of intracranial abnormality, whether the intracranial abnormality was an intraparenchymal lesion, presence of hemorrhage within the intraparenchymal lesion, supratentorial versus infratentorial location, side of intraparenchymal lesion, and presence of intraventricular hemorrhage (IVH). A positive score for ICH was determined by raters identifying an intracranial abnormality as an intraparenchymal lesion with hemorrhage in the correct location and anatomical side, in agreement with the available conventional radiologic report. The main outcome measure was the presence of ICH diagnosed on pMRI as determined by rater consensus.

**Intracerebral hemorrhage signal intensity ratios**. Low-field pMRI exams with a clinical diagnosis of ICH as confirmed by baseline standard-of-care neuroimaging were provided to four neuroimaging research core lab readers (M.H.M., M.M.Y.,

A.M.P., I.R.C.). Lesion annotation was completed manually by readers using Horos (v3.3.5) on T2W and FLAIR sequences. Readers annotated the hemorrhage core, hemorrhage rim (if distinguishable), and contralateral hemisphere on a single representative slice. Hemorrhage segmentations were used to compute signal intensity ratios (SIR).

**Hematoma volume, localization, and patient outcome.** Hematoma volume was measured by manual segmentation by one reader (M.H.M.) in AFNI (v21.1.02) and estimated using the ABC/2 method[44] by four neuroimaging research core lab readers (M.H.M., M.M.Y., A.M.P., I.R.C.) using Horos (v3.3.5). Raters annotated pMRI T2W and FLAIR sequences and the closet conventional imaging study (CT or 1.5/3T MRI T2W and FLAIR sequences) within 36 h of the pMRI exam. Low-field pMRI exams whose closest conventional imaging study occurred outside 36 h were excluded from this sub-analysis to account for the highly dynamic nature of ICH[56,57]. For the ABC/2 method, lesions with non-ellipsoid morphologies were excluded due to the inaccuracy of the ABC/2 measurement in these cases[45]. To localize lesions between pMRI and conventional imaging, point regions of interest (ROIs) were annotated by one reader (A.L.C.) in AFNI (v21.1.02) at the optic chiasm, septum pellucidum, and on the anatomical right and left side of the central sulcus at the level of the topmost slice of the lateral ventricles by drawing a 2-dimensional circle with a radius of 3 mm. The centroids of the manually segmented hematoma volumes and the point ROIs at each aforementioned anatomical location were computed in AFNI (v21.1.02) and used to calculate the Euclidean distances between the lesions and point ROIs. Cognitive scores were collected from the patients' electronic medical record and comprised of the closest National Institutes of Health Stroke Scale (NIHSS) obtained at the time of pMRI examination and patient functional outcome at discharge as assessed by the Modified Rankin Scale (mRS).

**Statistical methods.** Descriptive statistics are reported using means (SD) and medians (interquartile range [IQR]), as appropriate. Point estimates and 95% confidence intervals were determined for sensitivity, specificity, positive predictive value (PPV), and negative predictive value (NPV). Interrater reliability for sensitivity and specificity was calculated for observations of both T2W and FLAIR exams using the Gwet's AC2 agreement coefficient[59,60]. SIR were computed by dividing the mean signal intensity of the annotated lesion (hemorrhage core and, if applicable, hemorrhage rim) on a single representative slice by the mean signal intensity of the contralateral hemisphere. Averaged SIR and standard deviations were computed across individual raters. $SIR < 0.95$ were characterized as hypointense, $0.95 \leq SIR \leq 1.05$ were characterized as isointense, and $SIR > 1.05$ were characterized as hyperintense. For lesion localization across modalities, the Euclidean distance between the centroid of the segmented hemorrhage and each point ROI was calculated in RStudio v3.6.1 for each pMRI exam. The intraclass correlation coefficient (ICC) was used to assess the accuracy of pMRI hematoma volume measurements and Euclidean distances on T2W and FLAIR sequences compared to the closest conventional neuroimaging study measurements (CT or 1.5/3T MRI T2W and FLAIR). In addition, the ICC was used to assess interrater agreement between readers using the ABC/2 method to estimate hematoma volume. Manual and ABC/2 hematoma volumes on pMRI sequences (T2W and FLAIR) were correlated with National Institutes of Health Stroke Scale (NIHSS) obtained at the closest time to pMRI examination and Modified Rankin Scale (mRS) at discharge using unadjusted and adjusted linear regression models. Adjusted linear regression models included sex, race, and age. All statistical computations were completed using RStudio v3.6.1.

**Reporting summary.** Further information on research design is available in the Nature Research Reporting Summary linked to this article.

## Data availability

Source data are provided with this paper as a source data file. Raw data associated with Tables 1, 2, and 3 and Figs. 4 and 5 are included in the source data file. While a public portable MRI neuroimaging repository is not established yet, there are ongoing efforts to make the neuroimaging data publicly available. In the interim, the neuroimaging studies analyzed during the current study are available from the corresponding author upon request. Source data are provided with this paper.

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

## Acknowledgements

This study was supported by funding from the American Heart Association (Collaborative Science Award 17CSA3355004, co-PI KS, WTK, and MSR). The prototype device that was developed, deployed and provided to Yale was borne out of an academic-industry collaboration, where the AHA grant supported the academic investigators. KNS is supported by the NIH (U24NS107136, U24NS107215, R01NR018335, R01NS110721, R03NS112859, U01NS106513, and 1U01NS106513-01A1) and the American Heart Association (18TPA34170180 and 17CSA33550004) and a Hyperfine Research, Inc. research grant.

## Author contributions

The senior corresponding author (Dr. Kevin N. Sheth) takes responsibility for all independent decisions made regarding study concept, analysis, and conclusions. Study design, data interpretation, and writing of the manuscript: M.H.M., W.T.K., and K.N.S. Data analysis: M.H.M. Data acquisition: M.H.M., B.A.C., M.M.Y., A.M.P., I.R.C., J.T.S., E.B.W., C.W., A.W., and N.T. Contributions and critical revision to the manuscript: B.A.C., M.M.Y., A.M.P., I.R.C., A.L.C., E.B.W., C.W., C.C.M., A.L., R.B., T.J.P., J.W., J.P.A., G.J.F., K.T.G., N.P., J.S., L.S., E.J.G., D.Y.H., J.A.K., A.M., and G.S. Developed technology for this study: E.B.W., J.R., L.S., M.P., and M.S.R.

## Competing interests

K.N.S. is the principal investigator. This study received support from the Collaborative Science Award from the American Heart Association (PIs: K.N.S., W.T.K., M.S.R.), National Institutes of Health Supplement Grant, and Hyperfine Research, Inc. research grant. W.T.K. receives grants from NIH and AHA; grants and personal fees from Biogen, Inc; grants and personal fees from NControl Therapeutics; has a patent pending that is licensed to NControl Therapeutics; holds equity in Woolsey Pharmaceuticals. M.S.R. is a co-founder of Hyperfine Research, Inc. J.R. is a co-founder of Hyperfine Research, Inc. E.B.W., L.S., and M.P. are research scientists and engineers at Hyperfine Research, Inc. All other authors declare no competing interests.
