## [Peer Review File · Nature Communications]

Portable, Bedside, Low-Field Magnetic Resonance Imaging for Evaluation of Intracerebral HemorrhageReviewers' comments:

Reviewer #1 (Remarks to the Author):

General Comments

This is a timely study with an interesting approach (bedside MRI at ULF), although not new. Particularly to avoid extra patient transport and imaging in a strictly controlled environment of conventional MRI units may pay off for both, patient and hospital. This should also reduce problems with claustrophobia. In addition, ULF-MRI likely could be combined with EEG or MEG at bedside. I agree that the technique has the potential to provide diagnostic images for patients in critical care units, however, this study cannot yet provide a clinical proof for the "reliable and accurate for detection and characterization of ICH across a range of brain hemorrhage presentations" as compared to conventional high-field MRI.

Abstracts/Background

The authors state that conventional MRI operates at high (1.5-3 T) field. Although they are correct that > 80% of all clinical imagers operate at this field strength (Moser et al. 2012), particularly in critical care units mid-field systems (0.2 - 1 T) are employed for decades (e.g., Campbell-Washburn et al. 2019; Klein 2016), even for stroke (Bhat et al. 2020). Of course, these are conventional, large bore MRI systems and not bedside units. Nevertheless, the authors should mention this in the Abstract and Background. Furthermore, magnet field strengths in MRI below 0.2 T are commonly called low or ultra-low field (ULF). Noteworthy, Lothar et al already presented a mobile and efficient ULF (23 mT) MRI for neonatal applications in 2016. See missing refs.

Methods

The authors should add city and country of the portable, bedside MRI used in this study. What was the ramp time and linearity of the gradient system? How were the pulse sequences, installed by the manufacturer, optimized for this particular purpose? Why not using a spin-echo sequence with (much) lower TE to save SNR, measurement time, and reduce susceptibility artifacts? Any quality assurance (QA) protocol used to check the specs given by the manufacturer? As the scanning time was rather long, due to the low field and sensitivity, did the authors correct for motion artifacts which are rather common in ICH patients?

Results

First of all, sensitivity and specificity of the mobile MRI system, and pulse sequences, used should be given or, at least, a published reference (cf. Lothar et al. 2016).

The rate of successfully performed scans (68 %) seems rather low and should be compared to available numbers obtained in conventional MRI scanners.

The above may be a consequence of using such a long TE (172 ms and 252 ms, respectively), in addition to the very low field. Together, the imaging sequences used are definitely prone to a mixture of susceptibility and motion artifacts, particularly relevant in hemorrhage imaging (cf. figs. 2 and 3), pls. quantify and discuss.

Sensitivity of the diagnostic imaging results is not very impressive compared to conventional MRI, probably due to insufficient optimization of the hardware and pulse sequence parameters, and missing QA protocols.

Discussion

I do not agree with the statement that "Brain images acquired from this approach produced neuroimaging results that were reliable and accurate for detection and characterization of ICH across a range of brain hemorrhage presentations. Further, additional validation of this approach stems from our observations that ICH volume is associated with severity and outcomes, a cornerstone of clinical assessment. In the context of a portable solution, these results suggest that pMRI based assessments of ICH may be useful in a broader range of clinical settings and may enable widespread evaluation through this approach."

This would definitely require improved hardware and specifically optimized measurement protocols, including artifact correction.

Also, sensors to detect signals from the human brain using ULF-MRI could be more sensitive than a simple 8-channel head coil (Q-factor?) I would say.

Conclusion

Premature (see other comments)

Missing references

Bhat, S.S., Fernandes, T.T., Poojar, P., da Silva Ferreira, M., Rao, P.C., Hanumantharaju, M.C., Ogbole, G., Nunes, R.G. and Geethanath, S. Low-Field MRI of Stroke: Challenges and Opportunities. *J Magn Reson Imaging*. (2020), EPub August 22, 2020. doi:10.1002/jmri.27324

Campbell-Washburn AE, Ramasawmy R, Restivo MC, Bhattacharya I, Basar B, Herzka DA, et al. Opportunities in interventional and diagnostic imaging by using high-performance low-field-strength MRI. *Radiology*. (2019) 293:384–93. doi: 10.1148/radiol.2019190452

Klein HM. *Clinical Low Field Strength Magnetic Resonance Imaging*. Cham: Springer International Publishing AG Switzerland (2016). doi: 10.1007/978-3-319-16516-5

Lothar S, Schiff SJ, Neuberger T, Jakob PM, Fidler F. Design of a mobile, homogeneous, and efficient electromagnet with a large field of view for neonatal low-field MRI. *Magn Reson Mater Phys*. (2016) 29:691–8. doi: 10.1007/s10334-016-0525-8

Moser E, Stahlberg F, Ladd ME, Trattnig S. 7-T MR--from research to clinical applications?. *NMR Biomed*. 2012;25(5):695-716. doi:10.1002/nbm.1794

Moser E, Laistler E, Schmitt F, Kontaxis G. Ultra-High Field NMR and MRI—The Role of Magnet Technology to Increase Sensitivity and Specificity. *Frontiers in Physics*. 2017;5:33. DOI 10.3389/fphy.2017.00033

Sarracanie M, Salameh N. Low-Field MRI: How Low Can We Go? A Fresh View on an Old Debate. *Frontiers in Physics*. 2020;8:172. DOI 10.3389/fphy.2020.00172

Reviewer #2 (Remarks to the Author):

The authors describe the application of a new, low-field (0.064 Tesla) MRI system for bedside imaging of patients suspected of intracranial hemorrhage. Specifically, the authors recruited 102 patients with intracranial hemorrhage (ICH) or acute ischemic stroke (AIS) for study, of which 68 were successfully imaged. The authors report sensitivity of 80% (32 of 40 known positive) and specificity of 100% (28 of 28 known negative). The authors describe potential advantages of the new approach and areas for improvement.

This work is one of, if not the first, description of the use of such a portable MRI system, and thus the presentation is of considerable interest to those working in the MRI field. The system has a high degree of novelty. The concept of a portable, readily-accessible MRI system will be interesting to many who work outside MRI.

This reviewer has several points of critique:

1. The review and determination of positive or negative findings need improved clarity. The work states that this was done by five neurologists. The authors should indicate the levels of experience of the readers in reviewing such images. Also, the authors should clearly state how a positive score was determined; e.g. did this require that all five readers indicate a suspected lesion in a specific region?
2. The authors should provide more technical information. Please explain in a sentence or two what a "biplanar" gradient system is and whether or how this may limit performance vs. gradients in three directions on current MRI systems. For example, are certain slice orientations not allowed? How many slices were acquired during the stated scan times? The two sequences were "pre-

configured." Can the authors explain briefly why these sequences (T2W and FLAIR) were selected?

3. The system is described as "portable." What is the elapsed time from when the system is brought into a room and when the actual MRI data acquisition can start for a subject in that room?

4. 15 patients were not imaged because of body habitus. Please elaborate. What is the lowest level in the head or neck which can typically be imaged?

5. A technical limitation appears to be distortion, as seen in several of the left and center column (pMRI) images in, e.g. Figs. 2A and 3B-C compared to the right column images. Please comment.

Other points of less importance are:

6. Abstract, first word. Suggest "Radiologic" as "radiographic" implies radiographs

7. P 7, midway. "compared to traditional high-field MRI or CT"

8. P 8. How were average conventional volumes determined? Related to this, Fig. 4A should show the line of identity, as two measures of the same quantity (volume) are being compared

9. Use of domain transform manifold learning seems highly speculative. If this statement is retained it should be identified as such.

10. Fig. 1 can be condensed, as there is a lot of unused white space.

Reviewer #3 (Remarks to the Author):

In this paper, the authors study the diagnostic characteristics of a portable MRI system in the neurocritical care setting.

Major concerns:

1. This article describes a very specialized use case of a very specialized instrument. Unless there is a broader application (like enabling MRI in the setting of typically-contraindicated implants) that has been studied and can be supported with additional data, a more subspecialty journal (like JAMA Neurology in which a very similar proof-of-concept paper has recently appeared) is probably more appropriate.

2. There is no explanation for the decision to conduct a subgroup analysis on supratentorial parenchymal hemorrhages only. The fact that they are the "most common" (in this case ~67%) of the hemorrhages is not sufficient to justify analyzing this group separately. Why not analyze posterior fossa hemorrhages since these are typically more life-threatening and important to recognize quickly in the neuroICU setting? Ideally, the decision to analyze subgroups of data should be based on a sound hypothesis developed prior to data collection/analysis; otherwise, this appears to serve no purpose other than to report a more impressive sensitivity number compared to the 80% sensitivity for all ICH. Also, the sensitivity of 80% really should not be characterized as "robust" in the discussion. (If a radiologist were overlooking 20% of ICH, his/her performance certainly would not be characterized as robust.)

3. The imaging really should also be scored by board-certified, CAQ neuroradiologists. Though hindsight always makes missed findings more obvious, several of the "false negatives" appear to reach RADPEER level 3 (should be identified most of the time).

4. Perhaps related to #3, but some discussion also needs to be made regarding the inter-rater reliability scores for ICH detection. While the p-values indicate that agreement is better than random chance, the k values reveal that the level of reliability between raters is only moderate (< 0.6). So it seems that the test performance depends substantially on the reader.

5. The term "prospective" appears in the manuscript multiple times and is confusing in this context. Typically, prospective is used to describe the timing of the selection of the cohort in a longitudinal observational study, which this is not. The current study investigation appears to be a review of imaging data that had already been acquired and used for clinical care.

Minor:

1. Though it may not be practical to perform a direct comparison, some mention should be made of portable CT systems (e.g. Ceratom) that solve many of the issues regarding patient transport.

2. Some acknowledgment should be made about the significant decreases in radiation dose that have been achieved with advanced reconstruction methods that are now standard on modern CT units, and how this compares to the potential benefits of rapid, accessible neuroimaging in the acute/critical setting.

3. First word in the abstract should be Radiological. Radiographic examination of the brain is rather limited.

Portable, Bedside, Low-Field Magnetic Resonance Imaging for Evaluation of Intracranial Hemorrhage

We are grateful to the Editors at *Nature Communications* for giving us the opportunity to submit a revised version of our manuscript, and to the Reviewers for their careful consideration. With their guidance, we have modified the initial submission to address issues raised in the review process. All editorial requirements have been satisfied in the revised submission. Please find below an itemized response to reviewers' comments and suggestions. We have made significant revisions to the methodology of this report in which ICH scoring and ascertainment were remodeled in order to establish a more rigorous and experienced panel of board-certified neuroradiologists. In addition, we have revised the scope and framing of this report to better reflect the potential benefits of the device as well as the obstacles that must be overcome.

Reviewer 1

- We are grateful for the comments from the first Reviewer, and we attempt to demonstrate responsiveness to each of the very helpful points.

Abstract/Background

The authors state that conventional MRI operates at high (1.5-3 T) field. Although they are correct that > 80% of all clinical imagers operate at this field strength (Moser et al. 2012), particularly in critical care units mid-field systems (0.2 - 1 T) are employed for decades (e.g., Campbell-Washburn et al. 2019; Klein 2016), even for stroke (Bhat et al. 2020). Of course, these are conventional, large bore MRI systems and not bedside units. Nevertheless, the authors should mention this in the Abstract and Background. Furthermore, magnet field strengths in MRI below 0.2 T are commonly called low or ultra-low field (ULF). Noteworthy, Lother et al already presented a mobile and efficient ULF (23 mT) MRI for neonatal applications in 2016. See missing refs.

- We appreciate these points and agree that there is more in the literature that should be included in the Background section of the paper. Additional references have been incorporated into the Background and Discussion, and we have edited the manuscript in order to provide further information regarding previous uses of mid- and low-field MRI technology.

Methods

The authors should add city and country of the portable, bedside MRI used in this study.

- The portable MRI device used in this study is from the Hyperfine Research Inc. medical equipment manufacturing site in Guilford, Connecticut. This information has been added to the manuscript's Methods section under "Portable MRI Specifications."

What was the ramp time and linearity of the gradient system?

- Taken from the Hyperfine scanner Instructions for Use, the peak amplitudes of the gradients are 26 mT/m (on Z-axis) and 25 mT/m (on X- and Y-axis). The Z-axis is the up-down direction between the magnet pole shoes (anterior-posterior for a patient lying inside the scanner). The gradient peak slew rates are 67 T/m/s (on Z-axis) and 23 T/m/s (on X- and Y-axis). Gradient linearity is not reported by Hyperfine. This information has

been added to the Methods section of the revised manuscript under “Portable MRI specifications.”

How were the pulse sequences, installed by the manufacturer, optimized for this particular purpose?

- Pulse sequences were not optimized for our study, but we agree with the Reviewer that this may be beneficial in the future in order to correct for motion artifact, achieve shorter acquisition times, and increase sensitivity so that the device can be expanded for broader clinical use. Each pMRI examination that we obtain is uploaded to the Hyperfine Cloud Picture Archive and Communication System and is reviewed by Hyperfine clinical scientists. As we perform more pMRI examinations, we make observations and collect feedback on how to best optimize the device in future hardware and software updates. As the device hardware and software as well as image reconstruction tools improve with time, we speculate that device detection and image quality could improve on the device.

Why not using a spin-echo sequence with (much) lower TE to save SNR, measurement time, and reduce susceptibility artifacts?

- We agree this is a valuable question. Due to the ultra-low field (ULF) strength of the device, a higher TE time is required in order to obtain adequate k-space at low magnetic field strength. Because of this, there is an inherent reduction in SNR, which consequently limits spatial resolution. In order to account for this, tradeoffs must be made in image acquisition time and presence of susceptibility artifacts. For this pMRI device, the TE was empirically determined to give sufficient cerebrospinal fluid contrast relative to gray and white matter. Over time, the device has undergone multiple software upgrades which have adjusted the TE time. We anticipate further improvements in software as well as image reconstruction tools will allow for adequate k-space to be obtained at ULF with a lower TE thereby saving SNR, measurement time, and reducing susceptibility artifacts. For clarity, a more detailed description of these limitations has been added to the Discussion section of the revised manuscript.

Any quality assurance (QA) protocol used to check the specs given by the manufacturer?

- Given our goal of reporting the diagnostic sensitivity of the imaging results, we appreciate that any information regarding QA protocol is valuable. The scanner comes with a system quality assurance (SQA) phantom recommended to be scanned by Hyperfine on a monthly basis. SQA image sets are uploaded to the Hyperfine Cloud PACS and monitored by Hyperfine. According to Hyperfine, each scanner passes a phantom-based factory acceptance test (FAT) before delivery. The FAT verifies performance metrics established according to NEMA standards and reported to the US FDA as part of Hyperfine's 510(k) FDA submissions. However, the details of the FAT testing are not shared publicly. We have included this information in the Methods section of the revised manuscript, under “Portable MRI Specifications.”

As the scanning time was rather long, due to the low field and sensitivity, did the authors correct for motion artifacts which are rather common in ICH patients?

- We agree with the Reviewer that the scanning times for portable (0.064T) MRI are long in comparison with conventional (1.5T or 3T) MRI evaluations. We have included Table

2 in the revised manuscript to demonstrate this comparison. While the low magnetic field strength allows for the device to operate in the presence of ferromagnetic materials, it increases the acquisition time. The T2W and FLAIR sequences include automatic post-processing methods such as geometric distortion correction and intensity correction. No other methods were used to correct for motion artifacts. However, we anticipate further improvements in image reconstruction tools that may reduce susceptibility to motion artifacts and help increase spatial resolution at low field. Further study will be required to determine whether these advances will help reduce motion artifacts.

Results

The rate of successfully performed scans (68 %) seems rather low and should be compared to available numbers obtained in conventional MRI scanners.

- Within the second draft of this manuscript, we have made significant revisions to the methodology, and the cohort provided to readers for adjudication was modified accordingly. Within the revised cohort, we attempted to scan 112 patients presenting with a diagnosis of ICH or AIS. Of these, eight patients terminated the exam early due to sudden onset of claustrophobia, pain in the back or neck, or nausea that precluded our ability to obtain a pMRI examination. Of the 119 pMRI examinations on 104 patients that we did obtain, nine patients had a body habitus that prevented full entry into the scanner's opening and produced an incomplete field-of-view and six exams were motion degraded. Overall, we had a successful scan acquisition rate of 93%, and a successful pMRI examination quality rate of 90%.

The above may be a consequence of using such a long TE (172 ms and 252 ms, respectively), in addition to the very low field. Together, the imaging sequences used are definitely prone to a mixture of susceptibility and motion artifacts, particularly relevant in hemorrhage imaging (cf. figs. 2 and 3), pls. quantify and discuss. Sensitivity of the diagnostic imaging results is not very impressive compared to conventional MRI, probably due to insufficient optimization of the hardware and pulse sequence parameters, and missing QA protocols.

- The Reviewer rightly points out a limitation of the acquired neuroimages. The TE used on the pMRI device was empirically determined to give sufficient cerebrospinal fluid contrast relative to gray and white matter. Due to the ultra-low field (ULF) strength of the device, a higher TE time is required in order to obtain adequate k-space at low magnetic field strength. In turn, tradeoffs were made in sequence acquisition time and sensitivity to susceptibility artifacts. Further improvements in software are currently being made to account for longer scan time and susceptibility and motion artifacts as well as advances in image reconstruction tools that will allow for adequate k-space to be obtained at ULF. We have edited the Discussion section to more appropriately describe how improvements in hardware and software, sequence parameters, and image reconstruction tools are necessary in order to achieve greater sensitivity and reduce susceptibility and motion artifact (as seen in parts of Figures 2 and 3). In addition, we have expanded the Methods section of the revised manuscript to discuss the QA protocols that Hyperfine Research, Inc. follows to ensure scan quality and inform further improvements.

Discussion

I do not agree with the statement that "Brain images acquired from this approach produced neuroimaging results that were reliable and accurate for detection and characterization of ICH"

across a range of brain hemorrhage presentations. Further, additional validation of this approach stems from our observations that ICH volume is associated with severity and outcomes, a cornerstone of clinical assessment. In the context of a portable solution, these results suggest that pMRI based assessments of ICH may be useful in a broader range of clinical settings and may enable widespread evaluation through this approach.”

This would definitely require improved hardware and specifically optimized measurement protocols, including artifact correction.

Also, sensors to detect signals from the human brain using ULF-MRI could be more sensitive than a simple 8-channel head coil (Q-factor?) I would say.

- The Reviewer’s point is well-taken, and we have adjusted the Discussion section to remove the emphasis on “enabling widespread evaluation.” Instead, we focus on the potential for this device to provide a portable solution which could be applied in different settings for diagnostic and monitoring purposes. We have also removed the phrase “reliable and accurate” with regards to the device’s detection and characterization capabilities. We have added a detailed description of the limitations of the Hyperfine device along with necessary improvements that must be made, which we speculate could enable the device to be used in a broad clinical setting. We believe this revised approach better reflects the potential benefits of the device as well as the obstacles that must be overcome.

Conclusion

Premature (see other comments)

- We agree that there are still hardware and measurement-protocol improvements that need to be done to further optimize this device for widespread clinical use. While the results in this series are preliminary, the data support the idea that pMRI is a safe and feasible neuroimaging solution that contributes valuable information in ICH evaluation. We have reworded the Discussion and Conclusion to reflect this so that we avoid making premature generalizations.

Missing references

Bhat, S.S., Fernandes, T.T., Poojar, P., da Silva Ferreira, M., Rao, P.C., Hanumantharaju, M.C., Ogbole, G., Nunes, R.G. and Geethanath, S. Low-Field MRI of Stroke: Challenges and Opportunities. J Magn Reson Imaging. (2020), EPub August 22, 2020. doi:10.1002/jmri.27324

Campbell-Washburn AE, Ramasawmy R, Restivo MC, Bhattacharya I, Basar B, Herzka DA, et al. Opportunities in interventional and diagnostic imaging by using high-performance low-field-strength MRI. Radiology. (2019) 293:384–93. doi: 10.1148/radiol.2019190452

Klein HM. Clinical Low Field Strength Magnetic Resonance Imaging. Cham: Springer International Publishing AG Switzerland (2016). doi: 10.1007/978-3-319-16516-5

Lothar S, Schiff SJ, Neuberger T, Jakob PM, Fidler F. Design of a mobile, homogeneous, and efficient electromagnet with a large field of view for neonatal low-field MRI. Magn Reson Mater Phys. (2016) 29:691–8. doi: 10.1007/s10334-016-0525-8

Moser E, Stahlberg F, Ladd ME, Trattnig S. 7-T MR--from research to clinical applications?. NMR Biomed. 2012;25(5):695-716. doi:10.1002/nbm.1794

Moser E, Laistler E, Schmitt F, Kontaxis G. Ultra-High Field NMR and MRI—The Role of Magnet Technology to Increase Sensitivity and Specificity. *Frontiers in Physics*. 2017;5:33. DOI 10.3389/fphy.2017.00033

Sarracanie M, Salameh N. Low-Field MRI: How Low Can We Go? A Fresh View on an Old Debate. *Frontiers in Physics*. 2020;8:172. DOI 10.3389/fphy.2020.00172

- We thank the Reviewer for this information, and additional references have been included in the Background and Discussion sections of the revised manuscript.

Reviewer 2

- We appreciate the reviewer comments, which we have incorporated in a revised submission of the manuscript. In particular, we note their characterization of the device as having a “high degree of novelty.” We also appreciate their statement that “the concept of a portable, readily-accessible MRI system will be interesting to many who work outside MRI.”

The review and determination of positive or negative findings need improved clarity. The work states that this was done by five neurologists. The authors should indicate the levels of experience of the readers in reviewing such images. Also, the authors should clearly state how a positive score was determined; e.g. did this require that all five readers indicate a suspected lesion in a specific region?

- We agree that this information should be clarified and expanded upon in the manuscript. We have revised the methodology of this manuscript to have CAQ neuroradiologists review the exams. One neuroradiologist (G.S.) has 40 years of experience and the other neuroradiologist (A.M.) has 22 years of experience. An ICH imaging core lab researcher (A.L.) with 7 years of experience reviewing clinical neuroimaging of ICH evaluated the cases of disagreement between neuroradiologists. We have updated the Methods section entitled “Sensitivity and Specificity Rater Evaluation” to include this information.
- The readers each reviewed the pMRI examinations and completed a survey for each exam. The following data was recorded by each rater for each exam within the survey: presence of intracranial abnormality, whether the intracranial abnormality was an intraparenchymal lesion, presence of hemorrhage within the intraparenchymal lesion, supratentorial versus infratentorial location, side of intraparenchymal lesion, and presence of intraventricular hemorrhage (IVH). A positive score for ICH was determined by raters identifying an intracranial abnormality as an intraparenchymal lesion with hemorrhage in the correct location and anatomical side. We have expanded the Methods section entitled “Sensitivity and Specificity Rater Evaluation” to include this information.

2. The authors should provide more technical information. Please explain in a sentence or two what a “biplanar” gradient system is and whether or how this may limit performance vs. gradients in three directions on current MRI systems. For example, are certain slice orientations not allowed? How many slices were acquired during the stated scan times? The two sequences

were “pre-configured.” Can the authors explain briefly why these sequences (T2W and FLAIR) were selected?

- We appreciate the Reviewer’s questions and agree that there should be more explanation regarding the device’s technical information in the paper. The description of “biplanar” gradients refers to the physical shape of the gradient coils inside the two flat planes of the magnet pole shoes. Similar to conventional fixed-location MRI scanners, the Hyperfine gradient system includes X, Y, and Z direction gradient coils that enable any 3D imaging orientation. Both the T2W and FLAIR sequences have 36 slices each. This information has been incorporated into the Methods section of the revised manuscript under “Portable MRI Specifications.”
- As it currently stands, the portable MRI device has five sequence options for imaging the brain: T2W, FLAIR, T1W, and DWI with ADC. However, the device has undergone numerous software updates since the study began in 2018. These updates added new sequences, new imaging planes (sagittal and coronal), and improved upon previous sequences. The T2W and FLAIR sequences were selected for evaluation in this study because they were the only two sequences that were consistently available on the device across software changes occurring from the beginning of the study (July 2018) to its completion (November 2020).

3. The system is described as “portable.” What is the elapsed time from when the system is brought into a room and when the actual MRI data acquisition can start for a subject in that room?

- We agree with the Reviewer that the elapsed time for pMRI setup and data acquisition is an important value to report. We prospectively timed five pMRI scan set-up times for non-intubated patients in the neuroscience intensive care unit at Yale New Haven Hospital. On average, the time it takes to prepare the patient’s hospital room for scanner entry (repositioning lines and moving patient’s bed to create clearance for the pMRI) took $01:28 \pm 0:02$ min:s. Moving the scanner from the hall and properly positioning at the head of the bed took $00:48 \pm 0:01$ min:s. Once the scanner was at the head of the patient’s bed, positioning the patient inside the head coil took $05:13 \pm 0:08$ min:s with 1-2 research associates and one nurse. Initiation of scan acquisition once the patient was positioned took $00:53 \pm 0:03$ min:s. Because of the ultra-low field magnet, there was safe entry into the clinical environment and ferromagnetic objects required for clinical care did not need to be removed for scanning. Clinical and research staff could remain in the patient’s hospital room during scan acquisition. Moreover, the scanner’s open geometry design allows for easy access to intravenous lines, ventilation tubing, and intraventricular drains. The low decibel range of the pMRI scanner (60-80 db) is safety within the recommended cutoff of 99 db for hearing protection, allowing safe entry into the hospital room without any necessary precautions.
- We have included Table 2, which demonstrates the average recorded times for pMRI scan preparation, sequence acquisition, and scan termination steps for five non-intubated patients in the neuroscience ICU. We compare these values to the average recorded times for scan preparation, sequence acquisition, and scan termination times on a conventional 3T MRI system (Siemens MAGNETOM Verio 3T eco MRI) and 1.5T MRI system (Siemens AVANTO 1.5T MRI) for five non-intubated NICU patients at Yale

New Haven Hospital. We recognize that the sequence acquisition time is longer for portable MRI examinations, but note the significant decrease in time required for scan preparation and scan termination. Furthermore, the pMRI examinations can take place at the point-of-care and the patient does not have to be disconnected from any monitors or ICU equipment for thereby limiting the potential for adverse events to occur outside the ICU.

4. 15 patients were not imaged because of body habitus. Please elaborate. What is the lowest level in the head or neck which can typically be imaged?

- We have revised the methodology of this manuscript to have a panel of CAQ neuroradiologists review the cases, and in turn, have revised the cohort given to readers for adjudication. The number of patients excluded due to body habitus is nine patients in this revised manuscript. To assess the lowest level in the head or neck which can typically be imaged, we analyzed all pMRI examinations included in this analysis. Of the 144 pMRI exams, 107 reached the level of the medulla, 25 reached the level of the pons, 7 reached the level of the lateral ventricles, and 5 reached the level of the midbrain. From this, we note that the majority of patients had a complete pMRI FOV that reached to the level of the medulla, allowing for full brain insertion and infratentorial evaluation.
- We have included this information in the Results section of this manuscript entitled “Patient Cohort and Safety” and also provided a note in the Discussion section in order to elaborate on why these nine patients were excluded due to body habitus. In addition, we have included Figure 1, which details the dimensions of the portable MRI device, noting the measurements that affect patient entry due to body habitus. In general, the patient is limited entry into the portable MRI due to at least one of the following: head size, chest height, neck length, and shoulder width. Patients with a head circumference less than 72.9 cm, a chest height less than 32 cm, and a shoulder width less than 55 cm can comfortably be positioned in the pMRI scanner. However, it varies depending on the specific body shape of the patient. In the future, it would be beneficial to conduct a systematic study to help us understand which populations can and cannot receive pMRI scans across a variety of applications.

A technical limitation appears to be distortion, as seen in several of the left and center column (pMRI) images in, e.g. Figs. 2A and 3B-C compared to the right column images. Please comment.

- The Reviewer’s note about distortion in parts of Figures 2 and 3 is well-taken. We have edited the Discussion section to more appropriately describe the limitations of the portable MRI device and the improvements that would be needed to increase sensitivity and reduce susceptibility and motion artifacts, as seen in these figures. We anticipate that improvements to hardware and software, optimization of pulse sequence parameters, and the incorporation of image reconstruction tools will be able to increase image quality. Nevertheless, further study will be needed to determine whether these improvements will improve sensitivity and reduce susceptibility and motion artifact.

Other points of less importance are:

Abstract, first word. Suggest “Radiologic” as “radiographic” implies radiographs.

- This sentence has been adjusted in the revised Abstract.

P 7, midway. “compared to traditional high-field MRI or CT”

- This sentence has been adjusted in the revised manuscript, and we thank the Reviewer for noticing this detail.

How were average conventional volumes determined? Related to this, Fig. 4A should show the line of identity, as two measures of the same quantity (volume) are being compared.

- Conventional volumes were manually measured by one rater using AFNI (v21.1.02) and estimated using the ABC/2 method by the four neuroimaging research core lab readers using Horos (v.3.3.5). The methods used for measuring hematoma volume on conventional imaging were identical to those used to determine hematoma volume on the portable MRI examinations. This detail has been added to the Methods section under “Hematoma Volume, Hematoma Localization, and Patient Outcome.”
- We appreciate and agree with the Reviewer’s suggestion that Figure 4A should show the line of identity, and we have updated the Figure accordingly.

Use of domain transform manifold learning seems highly speculative. If this statement is retained it should be identified as such.

- We agree with the Reviewer and have revised the Discussion section to reflect the speculative nature of the statement.

Fig. 1 can be condensed, as there is a lot of unused white space.

- Figure 1 has been removed from the revised submission of this manuscript.

Reviewer 3

- We appreciate the Reviewer’s helpful feedback and have incorporated changes in an effort to improve the overall manuscript.

Major concerns:

1. This article describes a very specialized use case of a very specialized instrument. Unless there is a broader application (like enabling MRI in the setting of typically-contraindicated implants) that has been studied and can be supported with additional data, a more subspecialty journal (like JAMA Neurology in which a very similar proof-of-concept paper has recently appeared) is probably more appropriate.

- The *JAMA Neurology* paper was widely viewed (Altmetric 863 and over 28K views) and is now being cited. However, the initial paper was purely descriptive. It provided no information on assessment of utility for any application or any validation data. Here, in evaluating stroke, we examine one of the most common indications for urgent head imaging in the inpatient setting – not just in neurology or emergency medicine, but across a health system and around the world. This will be the first paper with any systematic validation. And while

promising, the paper also quantifies limitations of this approach for this application. We believe these data will be of interest to a broad, general scientific and clinical audience.

There is no explanation for the decision to conduct a subgroup analysis on supratentorial parenchymal hemorrhages only. The fact that they are the "most common" (in this case ~67%) of the hemorrhages is not sufficient to justify analyzing this group separately. Why not analyze posterior fossa hemorrhages since these are typically more life-threatening and important to recognize quickly in the neuroICU setting? Ideally, the decision to analyze subgroups of data should be based on a sound hypothesis developed prior to data collection/analysis; otherwise, this appears to serve no purpose other than to report a more impressive sensitivity number compared to the 80% sensitivity for all ICH. Also, the sensitivity of 80% really should not be characterized as "robust" in the discussion. (If a radiologist were overlooking 20% of ICH, his/her performance certainly would not be characterized as robust.)

- The Reviewer's point is well-taken, and it was not our intention to present findings in such a way that would result in a more impressive sensitivity number. We have reworded the Results section so that we simply report the sensitivity of supratentorial ICH, rather than describing it as a "subgroup analysis". We felt that it was meaningful to specify the sensitivity for this subset of intracranial hemorrhage cases, given that this type occurs most commonly in patients admitted for ICH management. As represented by our systematic cohort, supratentorial ICH is more prevalent and as the Reviewer notes, ICH location is a standard approach for subgroup presentation. We are simply mirroring this convention, which we also believe is clinically relevant.
- We also appreciate the Reviewer's mention that posterior fossa hemorrhages are more life-threatening and have noted the improvements that need to be made for evaluating these in the Discussion section. Within this preliminary series, there were six posterior fossa hemorrhages and 50 hemorrhages in the supratentorial region. In turn, we avoid citing the sensitivity for strictly posterior fossa hemorrhages simply due to the low n of examined subjects. However, we anticipate scanning more patients with hemorrhages in the posterior fossa and agree that evaluating the sensitivity of these particular hemorrhages is an important and meaningful clinical question.
- Lastly, we fully agree with the Reviewer that a sensitivity of 80% should not be characterized as "robust" in the Discussion. We have since modified this word choice to properly reflect the sensitivity of 80.4% (as computed for the revised methodology of this manuscript) as "notable," noting that it shows the majority of ICH was detected. However, strides must be made in scanner hardware and software and image reconstruction tools that we speculate could improve image quality and ICH detection. To be clear, this degree of sensitivity for detection demonstrates promise as a screen but certainly does not cross the threshold for replacement over any existing technology (which is not the claim here). Nevertheless, whether detection can be most efficiently improved with hardware or software advances, image reconstruction tools, automated detection strategies, or reader training will require further study.

The imaging really should also be scored by board-certified, CAQ neuroradiologists. Though hindsight always makes missed findings more obvious, several of the "false negatives" appear to reach RADPEER level 3 (should be identified most of the time).

- We appreciate the Reviewer's recommendation and have completely modified the methodology of this analysis accordingly. We had a panel of two CAQ neuroradiologists (G.S. 40 years of experience, A.M. 22 years of experience) independently evaluate T2W

and FLAIR pMRI exams. In addition, an ICH imaging core lab researcher (A.L.) with 7 years of experience reviewing clinical neuroimaging of ICH evaluated the cases of disagreement between neuroradiologists. We have rewritten the Methods section of this manuscript to reflect the change in readers. These changes to the methodology are also reflected throughout the Results and Discussion sections of this manuscript.

Perhaps related to #3, but some discussion also needs to be made regarding the inter-rater reliability scores for ICH detection. While the p-values indicate that agreement is better than random chance, the k values reveal that the level of reliability between raters is only moderate (< 0.6). So it seems that the test performance depends substantially on the reader.

- We agree with the Reviewer that the moderate kappa value seems to indicate that the test performance was dependent on the reader. For this resubmission, we have modified the methodology of this manuscript to have CAQ neuroradiologists adjudicate the cases in lieu of the previous five neurologists. In doing, we computed new inter-rater reliability scores for ICH detection among the neuroradiologists. Furthermore, we report the inter-rater reliability using the Gwet AC2 statistic, which is a variation of the Gwet AC1 statistic that can handle more than two raters. The Gwet AC statistic is commonly cited as more reliable than Cohen's Kappa since it provides a more stable inter-rater reliability coefficient and is found to be less affected by prevalence and marginal probability.
- In the iteration of this manuscript, we cite an overall inter-rater reliability using Gwet's AC2 of 0.791 (95% CI: [0.718-0.864], p=0), which is considered to be "substantial" agreement among raters and the p-value indicates that agreement is better than random chance. We have included these data in the revised Results section of the manuscript.

5. The term "prospective" appears in the manuscript multiple times and is confusing in this context. Typically, prospective is used to describe the timing of the selection of the cohort in a longitudinal observational study, which this is not. The current study investigation appears to be a review of imaging data that had already been acquired and used for clinical care.

- We agree with the Reviewer that the term "prospective" is inappropriate for our purposes, and the manuscript has been edited to remove this term. This report is rather an observational analysis.

Minor:

1. Though it may not be practical to perform a direct comparison, some mention should be made of portable CT systems (e.g. Ceratom) that solve many of the issues regarding patient transport.

- We agree with the Reviewer that information regarding portable CT systems is important and should be included in this paper. We have expanded the Discussion section of this paper to discuss the literature on portable CT systems (Ceratom) in comparison to the portable MRI system.

2. Some acknowledgment should be made about the significant decreases in radiation dose that have been achieved with advanced reconstruction methods that are now standard on modern CT units, and how this compares to the potential benefits of rapid, accessible neuroimaging in the acute/critical setting.

- We appreciate the Reviewer's comment and have updated the Background section to acknowledge previous strategies which have been developed in order to decrease radiation burden associated with fixed-location CT units. However, we maintain that an MRI solution would be more beneficial in order to avoid any patient exposure to harmful radiation in addition to eliminating the need for lead shielding and radiation safety protocols for scanner operation. Furthermore, we note that the pMRI solution can be brought to the hospital beside. While portable CT devices have been introduced to allow for more rapid and accessible neuroimaging in intensive care settings, these devices still require trained technicians and lead shielding to operate whereas the pMRI can be utilized in the absence of an MRI technician and allows for unrestricted access into the hospital room during acquisition. We believe the lack of radiation burden and its corresponding shielding requirements coupled with the evaluative benefits of MRI in comparison to CT may encourage the use of pMRI units. We have included this information in the revised Discussion section of the manuscript.

3. First word in the abstract should be Radiological. Radiographic examination of the brain is rather limited.

- This sentence has been adjusted in the revised Abstract.

REVIEWERS' COMMENTS

Reviewer #1 (Remarks to the Author):

The authors adequately responded to my comments and critique. In my view, the paper has improved significantly.

Reviewer #2 (Remarks to the Author):

This is a revision of a work in which the authors describe the application of a new, low-field (0.064 Tesla) MRI system for bedside imaging of patients suspected of intracranial hemorrhage (ICH). The authors imaged a total of 144 subjects which included 56 patients with ICH, 48 with acute ischemic stroke (AIS), and 40 normals. Readers correctly identified ICH in 45 of 56 subjects. Additional analysis of the data is provided.

The authors have substantially responded to the points of critique made of the original submission by this reviewer as well as the other two reviewers. This work is one of, if not the first, description of the use of such a portable MRI system, and thus the presentation is of considerable interest to those working in the MRI field. The system has a high degree of novelty. The concept of a portable, readily-accessible MRI system will be interesting to many who work outside MRI.

This reviewer has several remaining points of critique which are all minor:

1. Abstract. Rather than "adjudicate" suggest "evaluate."
2. Abstract. Midway through: define AIS when used for the first time.
3. MRI technique. When first described in the second paragraph on page 6, the T2W and FLAIR sequences should be stated as two-dimensional multi-slice (which they presumably were). Then, at the end of the section the term "3D" should be removed (as this implies something other than 2D multi-slice).
4. P 7 | 2. Suggest "86 cm wide"
5. P7 para 2. Give city, state for Apple.
6. P 7 para 3. FLAIR acquisition also has an inversion time TI in addition to TE and TR. These values should be indicated for the three software versions.
7. P 12 paragraphs 3 and 4. The numbers in each paragraph sum to 54, not 56. Please correct or explain.
8. Figure 3 legend is inconsistent with the figure in that part (F) is alluded to. Please correct.

Portable, Bedside, Low-Field Magnetic Resonance Imaging for Evaluation of Intracerebral Hemorrhage

We are grateful to the Editors at *Nature Communications* for giving us the opportunity to publish a suitably revised version of our manuscript, and to the Reviewers for their contributions to the peer review of this work. With their guidance, we have modified the submission to address issues raised in the review process. All editorial requirements have been satisfied in the revised submission. Please find below an itemized response to reviewers' comments and suggestions.

REVIEWERS' COMMENTS

Reviewer #1 (Remarks to the Author):

The authors adequately responded to my comments and critique. In my view, the paper has improved significantly.

- We thank the Reviewer for their prior guidance and contributions, which has helped the work improve.

Reviewer #2 (Remarks to the Author):

This is a revision of a work in which the authors describe the application of a new, low-field (0.064 Tesla) MRI system for bedside imaging of patients suspected of intracranial hemorrhage (ICH). The authors imaged a total of 144 subjects which included 56 patients with ICH, 48 with acute ischemic stroke (AIS), and 40 normals. Readers correctly identified ICH in 45 of 56 subjects. Additional analysis of the data is provided.

The authors have substantially responded to the points of critique made of the original submission by this reviewer as well as the other two reviewers. This work is one of, if not the first, description of the use of such a portable MRI system, and thus the presentation is of considerable interest to those working in the MRI field. The system has a high degree of novelty. The concept of a portable, readily-accessible MRI system will be interesting to many who work outside MRI.

This reviewer has several remaining points of critique which are all minor:

1. Abstract. Rather than "adjudicate" suggest "evaluate."

- We have changed the wording accordingly.

2. Abstract. Midway through: define AIS when used for the first time.

- We have defined AIS in the abstract and thank the Reviewer for noting this detail.

3. MRI technique. When first described in the second paragraph on page 6, the T2W and FLAIR sequences should be stated as two-dimensional multi-slice (which they presumably were). Then, at the end of the section the term "3D" should be removed (as this implies something other than 2D multi-slice).

- We agree with the Reviewer's point and have modified the Methods section to state that the T2W and FLAIR sequences were "two-dimensional multi-slice" images and removed the incorrect term "3D" in this section.

4. P 7 | 2. Suggest "86 cm wide"

- This is a valuable addition to improve clarity, and we have included in the Methods section of the manuscript.

5. P7 para 2. Give city, state for Apple.

- We have included the city, state for Apple in the Methods section of the manuscript to read Cupertino, CA, USA.

6. P 7 para 3. FLAIR acquisition also has an inversion time TI in addition to TE and TR. These values should be indicated for the three software versions.

- We thank the Reviewer for noting this addition and have included the FLAIR inversion time TI for the three software versions in the Methods section of this manuscript.

7. P 12 paragraphs 3 and 4. The numbers in each paragraph sum to 54, not 56. Please correct or explain.

- We appreciate the Reviewer identifying these two missing cases and have updated the Results section of the revised manuscript to include these for the T2W exams. These two cases were excluded from the FLAIR exams since the lesion was not visualized to annotate. The exclusion of these two cases for the FLAIR exams has been written in the Results section of the manuscript.

8. Figure 3 legend is inconsistent with the figure in that part (F) is alluded to. Please correct.

- We thank the Reviewer for their careful inspection and have corrected the Figure 3 legend accordingly.